# CLIQUE NUMBER ESTIMATION VIA DIFFERENTIABLE FUNCTIONS OF ADJACENCY MATRIX PERMUTATIONS

**Indradyumna Roy**[*1]**, Eeshaan Jain**[*2]**, Soumen Chakrabarti**[1]**, Abir De**[1]
[1]IIT Bombay, [2]EPFL
`{indraroy15, soumen, abir}@cse.iitb.ac.in`, `eeshaan.jain@epfl.ch`

## ABSTRACT

Estimating the clique number in a graph is central to various applications, *e.g.*, community detection, graph retrieval, etc. Existing estimators often rely on non-differentiable combinatorial components. Here, we propose a full differentiable estimator for clique number estimation, which can be trained from distant supervision of clique numbers, rather than demonstrating actual cliques. Our key insight is a formulation of the maximum clique problem (MCP) as a maximization of the size of fully dense square submatrix, within a suitably row-column-permuted adjacency matrix. We design a differentiable mechanism to search for permutations that lead to the discovery of such dense blocks. However, the optimal permutation is not unique, which leads to the learning of spurious permutations. To tackle this problem, we view the MCP problem as a sequence of subgraph matching tasks, each detecting progressively larger cliques in a nested manner. This allows effective navigation through suitable node permutations. These steps result in MxNet, an end-to-end differentiable model, which learns to predict clique number without explicit clique demonstrations, with the added benefit of interpretability. Experiments on eight datasets show the superior accuracy of our approach. The code is available on ⌂ GitHub.

## 1 INTRODUCTION

The Maximum Clique problem (MCP) entails finding the largest subgraph where every pair of nodes is connected by an edge. MCP is extensively studied among NP-Complete combinatorial problems (Karp, 2010), with efforts on designing exact and heuristic solvers for general and specialized graph classes (Reba et al., 2021; Carraghan & Pardalos, 1990; San Segundo et al., 2016; Depolli et al., 2013; Jiang et al., 2017; Tomita et al., 2017; Li & Quan, 2010; Konc & Janezic, 2007; McCreesh & Prosser, 2014). Beyond such theoretical appeal, MCP also has numerous real-world applications across molecular biology (Malod-Dognin et al., 2010; Depolli et al., 2013), social network analysis (Balasundaram et al., 2011), protein matching (Malod-Dognin et al., 2010; Ehrlich & Rarey, 2011; Depolli et al., 2013), identifying chemical reaction sites (Raymond & Willett, 2002; Agrafiotis et al., 2007; Fooshee et al., 2013), drug discovery (Cao et al., 2008), etc.

**Clique number estimation vs. MCP** Several of aforementioned applications require an accurate estimation of the size of the maximum clique, known as clique number, with less emphasis in predicting the maximum clique itself. This specifically include domains necessitating the comparison of graph structures, such as protein matching (Malod-Dognin et al., 2010; Ehrlich & Rarey, 2011; Depolli et al., 2013), reaction site identification (Raymond & Willett, 2002; Agrafiotis et al., 2007; Fooshee et al., 2013), drug discovery (Cao et al., 2008), etc. In such cases, the maximum common induced subgraph often serves as a relevance metric, directly mapped to the clique number detection of the modular product graph (Solnon et al., 2015). Various network analysis models study clique relations (Pattillo et al., 2013) for identifying cohesive subgroups in social networks (Balasundaram et al., 2011). In such applications, a key question is the decision version of MCP, *i.e.*, whether there is a clique of size $c$ exists in the graph, rather than finding the maximum clique. In such situations, where relevance is latently influenced by the clique size, a neural architecture capable of directly predicting clique number while ensuring end-to-end differentiability becomes essential.

**Prior work on MCP and their challenges in clique number estimation** With the proliferation of real-world applications, a series of recent neural solvers aim to solve MCP harnessing data distri-

---

*Indradyumna and Eeshaan contributed equally. Eeshaan Jain did this work while he was affiliated with IIT Bombay.

butional characteristics across supervised (Sun & Yang, 2024), unsupervised (Karalias et al., 2022; Karalias & Loukas, 2020; Min et al., 2022) and reinforcement learning (Sanokowski et al., 2023; Zhang et al., 2024) paradigms. However, they suffer from two key limitations, which prevent them from realizing their full potential in clique number estimation task.

**Lack of end-to-end differentiability:** End-to-end differentiability is a critical requirement for applications like graph matching, where the goal is quickly retrieve similar graphs from large corpus. Differentiable models enable us quick off-the-shelf use of indexing methods. However, most existing neural designs operate in conjunction with non-differentiable decoders, which impede their integration to such applications.

**Middle ground between extreme and no supervision:** Existing clique number estimation methods fall in two broad regimes. Combinatorial methods often use no training, whereas ML-based methods (Sun & Yang, 2024) depend on a full demonstration of an actual clique. We focus on the neglected middle ground, where supervision is provided as only the clique number (which can be estimated from various efficient relaxations) and not a demonstration (which is computationally expensive to collect). In several graph search and retrieval scenarios, the relevance of a corpus graph depends on the extent to which query and corpus graphs overlap, which can be reduced to a maximum common induced subgraph *size* computation.

## 1.1 OUR CONTRIBUTIONS

We introduce MxNet, a differentiable neural model specifically for clique number estimation rather than the maximum clique itself, by training under the distant supervision of the clique size without explicit clique guidance.

**Clique detection via message passing on permuted adjacency matrix** Suppose the input graph $G$ has a clique of size $c$, and to each node is added a self-loop. Then there exists a permutation, which, when applied on node indices, will let us write the node-node adjacency matrix in a manner that will reveal a $c \times c$ square block somewhere on the diagonal, completely filled with $c^2$ ones. If $\omega(G)$ is the clique number, the largest such square we can find will be $\omega(G) \times \omega(G)$. To turn this initial intuition into a neural estimator of $\omega(G)$, we need two differentiable gadgets: a network that proposes a node permutation, and a network that detects completely filled squares (maximal subsquare or MSS detection).

The latter problem admits a simple polynomial-time dynamic programming-type algorithm, which we generalize into a message-passing network over a grid-graph formed from the permuted adjacency matrix. This effectively transfers the computational complexity of the clique problem to the row-column permutation proposer. We devise an end-to-end differentiable network for this former combinatorial task by adapting a Gumbel-Sinkhorn network to propose a permutation that lets us discover large filled squares. Here, we naturally work in a relaxed continuous domain, and directly output an estimate of $\omega(G)$. We call this neural component of our model MxNet (MSS).

**Curriculum-matching a series of cliques** At least $|V(G)| \, \omega(G)!$ permutations (and possibly many more) can identify the MSS, leading to a huge multiplicity of global optima. These will muddle gradient signals, potentially damaging learning quality. During inference, even if $\omega(G)$ is estimated reasonably well, we may not get an interpretable permutation. In response, we design MxNet (SubMatch), the second variant of MxNet. MxNet (SubMatch) proposes a cascade of ever-larger cliques $\mathcal{K}_c$, $c = 2, 3, \ldots$, and prompts the permutation generator network to find an injective alignment of nodes in $\mathcal{K}_c$ to nodes of $G$. We expect these attempts to succeed as long as $c \leq \omega(G)$, and noticeably fail when $c > \omega(G)$. This expected behavior is worked into a suitable loss function. As we shall see, the inductive bias asserted through this 'curriculum' not only improves interpretability, but can also improve estimates of $\omega(G)$.

**End-to-end training and constrained early stopping.** We combine MxNet (MSS) and MxNet (SubMatch), described above, into MxNet (Composite), part of an end-to-end trainable framework we call MxNet. To mitigate overfitting and the learning of spurious correlations, we introduce a bicriteria early stopping method which effectively balances both components, ensuring accurate predictions supported by interpretable clique-based justifications. Our experiments on eight datasets show that MxNet offers significant accuracy boost beyond several baselines.

## 1.2 BRIEF DISCUSSION ON RELATED WORK

Apart from the aforementioned prior work on MCP, our work is also related to recent advancements in neural models for combinatorial optimization approaches on graphs and traditional methods for

estimating clique numbers in graphs. We briefly discuss them here. We further discuss related work in Appendix C.

**Neural models for combinatorial optimization on graphs**  In recent years, there is a series of works focusing on combinatorial optimization on graphs. This include designing neural networks which can solve graph matching (Yu et al., 2023; Liu et al., 2023; Wang et al., 2023; Nguyen et al., 2023; Zhou et al., 2023; Jiang et al., 2019; Yu et al., 2020; Fey et al., 2020). These works predominantly have applications in image matching. Several works design generic neural networks, which can solve a series of combinatorial optimization problems (Wang & Li, 2023; Zhao et al., 2023; Li et al., 2018; Goshvadi et al., 2023; Lu et al., 2023; Sun & Yang, 2024; Sanokowski et al., 2023; Zhang et al., 2024). Other works are tailored to solve specific problems including maximum clique (Karalias & Loukas, 2020; Min et al., 2022), maximum independent set (Brusca et al., 2023; Ahn et al., 2020), maximum common induced subgraph (Bai et al., 2021), etc.

**Clique number estimation without maximum clique computation**  Presenting the learner with $G$ and its clique number $\omega(G)$, rather than a maximal clique itself, presents a practical intermediate path between full supervision and training-free methods. Even though demonstrating $G$'s cliques to the learner is itself computationally intractable, $\omega(G)$ can be bounded (Pardalos & Phillips, 1990; Gibbons et al., 1997) via the optimal values of various continuous optimizations, the most well-known being $\max_{\boldsymbol{x}:\boldsymbol{x}\geq\boldsymbol{0},\boldsymbol{1}\cdot\boldsymbol{x}=1}\boldsymbol{x}^\top\boldsymbol{A}\boldsymbol{x}$, originally due to Motzkin & Straus (1965). Sharper bounds have been developed (Budinich, 2003). Pelillo (1995) demonstrated how such quadratic programs can be solved by message passing networks.

## 2  NOTATION AND PROBLEM SETUP

**Notation**  Given a graph $G = (V, E)$, we denote its adjacency matrix by $\boldsymbol{A}$, which is possibly padded with zero rows and columns to ensure all matrices $\boldsymbol{A}$ have the same dimension $N \times N$. We define $\Pi_N$ as the set of all $N \times N$ hard permutation matrices, and $\mathcal{P}_N$ as the set of all $N \times N$ soft permutation matrices, which are essentially doubly stochastic matrices. We will frequently use $\boldsymbol{P} \in \Pi_N$ ($\boldsymbol{S} \in \mathcal{P}_N$) to denote hard (respectively, soft) permutation matrices. We write $\mathcal{O}_c = \boldsymbol{1}_c\boldsymbol{1}_c^\top$, where $\boldsymbol{1}_c$ is an all-one vector of dimension $c$. $[a]_+ = \max\{a, 0\}$ represents the hinge or ReLU function. For a matrix $\boldsymbol{R}$, we denote $\sum_{u,v}|\boldsymbol{R}[u,v]|$ by $\|\boldsymbol{R}\|_{1,1}$. Given a binary matrix $\boldsymbol{B}$, we refer to the size of (number of rows or columns) of the maximum sized all-ones square submatrix as $\mathrm{MSS}(\boldsymbol{B})$. We call such "all-ones square submatrix" as "fully dense subsquare", to generalize to continuous $\boldsymbol{B}$.

**Maximum clique problem (MCP), clique number**  A clique $\mathcal{K}_c$ is a complete graph with $c$ number of nodes. The maximum clique $\mathcal{K}^*(G)$ in a graph $G$ is the clique within $G$, containing the highest number of nodes. The size of this maximum clique, *i.e.*, $\omega(G) = |V(\mathcal{K}^*(G))|$, is referred to as the *clique number*. The clique number $\omega(G)$ of a graph is unique, although the maximum clique itself may not be — there can be multiple maximum cliques $\mathcal{K}^*$ with same number of nodes $\omega(G)$. However, as discussed in several classic papers from the Operations Research literature (Bomze et al., 1999; Wilf, 1967; 1986), the clique number $\omega(G)$ can be bounded using the spectral properties of the graph $G$. When the computation of the maximum clique size is particularly challenging, these bounds can provide an estimation of the clique number, serving as distant supervisory labels for neural max-clique methods.

**Problem statement**  Given a graph $G$, our objective is to design an end-to-end differentiable neural model which takes the graph as input and outputs an estimate of its clique number. Formally, given a set of $I$ training instances $D = \{G_i, \omega(G_i) \,|\, i \in [I]\}$, we aim to train our model on $D$, so that it can predict the clique number of a new graph $G$. Our goal is to avoid any non-differentiable combinatorial routines during both training and testing phases. Our focus is on estimating specifically the clique number, rather than the clique itself, motivated by several real applications (Malod-Dognin et al., 2010; Ehrlich & Rarey, 2011; Depolli et al., 2013; Raymond & Willett, 2002; Agrafiotis et al., 2007; Fooshee et al., 2013; Cao et al., 2008).

## 3  THE DESIGN OF MXNET

We describe our proposed framework, MXNET, in three stages.

**MXNET (MSS)**  First, we describe how to combine a soft permutation generator with a network that searches for the largest square submatrix in a given matrix, to directly predict clique number.

**MXNET (SubMatch)**  While MXNET (MSS) gives accurate estimates of $\omega(G)$, the very large number of loss optima prevents it from finding sharply interpretable clique demonstrations. We

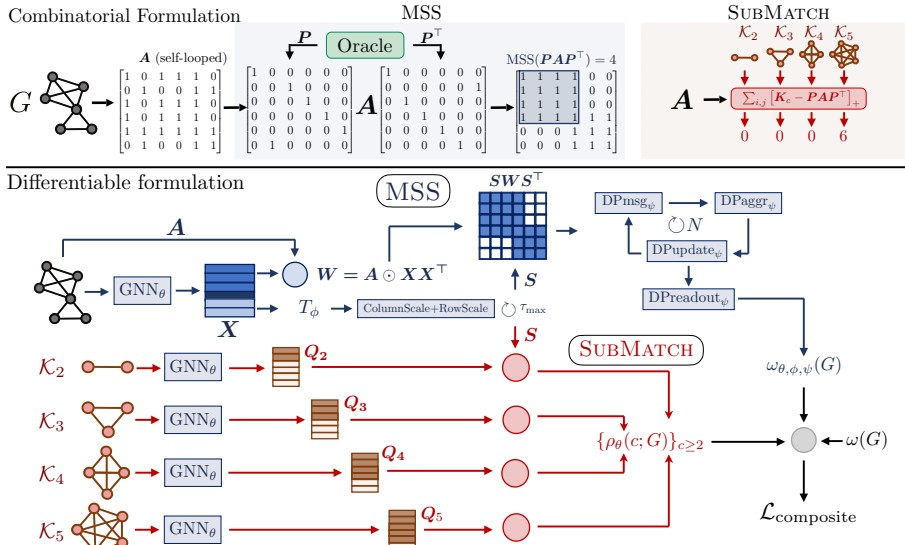

Figure 1: Input graph $G$ with adjacency matrix $\boldsymbol{A}$ contains $\mathcal{K}_4$ and $\mathcal{K}_3$, so $\omega(G) = 4$. The topmost part shows how a node permutation $\boldsymbol{P}$ supplied by an oracle rearranges $\boldsymbol{A}$ to locate the maximal clique as a dense square diagonal block. The lower part shows the two neural components that turn the combinatorial intuition into a differentiable network. In the first neural component (middle), $\text{GNN}_\theta$ obtains contextual node embeddings $\boldsymbol{X}$, which drives a Gumbel-Sinkhorn network $T_\phi$ to propose a soft permutation $\boldsymbol{S}$. $\boldsymbol{S}$ is applied to $\boldsymbol{W} = \boldsymbol{A} \odot \boldsymbol{X}\boldsymbol{X}^\top$ (rationale explained in text) to form the relaxed counterpart of $\boldsymbol{P}\boldsymbol{A}\boldsymbol{P}^\top$ in the oracular setting. If $\boldsymbol{S}$ is chosen well, we expect $\boldsymbol{S}\boldsymbol{W}\boldsymbol{S}^\top$ to have large, fully-filled diagonal squares. The largest among these is/are detected by message-passing ($\text{DPmsg}_\psi, \text{DPaggr}_\psi, \text{DPreadout}_\psi$) in a grid graph. The second neural component (shown at the bottom) consists of a series of cliques $\mathcal{K}_2 \subset \mathcal{K}_3 \subset \cdots$, each processed by $\text{GNN}_\theta$, to provide contextual node embeddings $\boldsymbol{Q}_2, \boldsymbol{Q}_3, \ldots$, which are checked for a form of injective subsumption by $\boldsymbol{X}$. Subsumption fails as we move from $\boldsymbol{Q}_{\omega(G)}$ to $\boldsymbol{Q}_{\omega(G)+1}$ — a transition we detect via the $\rho_\theta(c; G)$ network. The losses from the two neural components are combined into $\mathcal{L}_{\text{Composite}}$.

introduce a related strategy MxNet (SubMatch), where we prepare a series of cliques of increasing size and try to detect these cliques as subgraphs of the input graph. By construction, MxNet (SubMatch) is better at producing clique certificates.

**MxNet (Composite)** We combine MxNet (MSS) and MxNet (SubMatch) to build an accurate, yet interpretable clique number estimator MxNet (Composite). Here, MxNet (SubMatch) helps MxNet (MSS) navigate through suitable node permutations, while MxNet (MSS) promotes accuracy by directly predicting the clique number.

### 3.1 Design of MxNet (MSS)

We first introduce a novel combinatorial formulation of the Maximum Clique Problem (MCP) and then customize this formulation to develop MxNet (MSS), which approximates this combinatorial solution in a differentiable manner.

**Combinatorial formulation of MCP** Given a graph $G = (V, E)$, suppose the nodes are magically ordered in such a way that all those nodes that belong to one of the maximal cliques, are consecutively numbered. Under this node ordering, the submatrix indexed by $V(\mathcal{K}^*(G))$ in the corresponding adjacency matrix $\boldsymbol{A}$ (with added self loops) will be the largest possible fully dense (all-ones) subsquare in $\boldsymbol{A}$. Formally, if $V = [N]$ and $V(\mathcal{K}^*(G)) = \{j + 1, ..., j + \omega(G) \mid \text{for some } 1 \leq j \leq N - \omega(G)\}$, then we have (i) $\boldsymbol{A}[V(\mathcal{K}^*(G)), V(\mathcal{K}^*(G))] = \mathcal{O}_{\omega(G)}$ and (ii) for any pair $(j, j + c)$ with $c > \omega(G)$ and $1 \leq j \leq \omega(G) - c$, we have $\boldsymbol{A}[j : j + c, j : j + c] \neq \mathcal{O}_c$.

Given the above ordering among the nodes, the MCP is equivalent to finding the largest fully dense subsquare within the adjacency matrix, which can be solved in polynomial time. Therefore, all that we have achieved thus far is to relegate the hardness of MCP to the module that must suggest the node ordering. To more forward, we cast the MCP as a maximization of the size (number of rows or columns) of the largest fully dense subsquare in the adjacency matrix, over all possible node

permutations as the optimization variable. If we define $\mathrm{MSS}(\boldsymbol{B})$ as the size of the largest fully dense diagonal subsquare in a matrix $\boldsymbol{B}$, then MCP amounts to

$$\text{maximize}_{\boldsymbol{P} \in \Pi_N} \ \mathrm{MSS}(\boldsymbol{P}\boldsymbol{A}\boldsymbol{P}^\top). \tag{1}$$

The routine $\mathrm{MSS}(\cdot)$ is summarized in Algorithm 1. A matrix called FF (mnemonic for flood-fill, or frontier-flooding), which is updated using a simple dynamic programming (DP) routine. After updating the $(i,j)$-th position, $\mathrm{FF}[i,j]$ represents the size of the largest fully dense subsquare in $\boldsymbol{B}$ (which is $\boldsymbol{P}\boldsymbol{A}\boldsymbol{P}^\top$ in our problem), with the bottom-right element as $\boldsymbol{B}[i,j]$. Algorithm 1 first initializes the first row and column of FF with those from $\boldsymbol{B}$.

Once we update $\mathrm{FF}[r,c]$ with $r < i, c < j$, then $\mathrm{FF}[i-1,j]$, $\mathrm{FF}[i,j-1]$ and $\mathrm{FF}[i-1,j-1]$ contain the values of the size of largest fully dense subsquare in $\boldsymbol{B}$, with bottom right position as $(i-1,j), (i,j-1)$ and $(i-1,j-1)$ respectively. If $\boldsymbol{B}[i,j]=1$, the size of the corresponding subsquare ending at $(i,j)$ is computed by increasing one over the minimum of the above three entries of FF as: $\mathrm{FF}[i,j] = 1 + \min\{\mathrm{FF}[i-1,j], \mathrm{FF}[i,j-1], \mathrm{FF}[i-1,j-1]\}$. We can visualize this computation as a wavefront progressing down the main diagonal. Finally, we compute the required size of fully dense subsquare as $\max_{i \in [N]} \mathrm{FF}[i,i]$.

---

**Algorithm 1** $\mathrm{MSS}(\boldsymbol{B})$    # $\boldsymbol{B}$ is binary

---

1: $\mathrm{FF} \leftarrow \mathrm{Empty}(N,N)$
2: $\mathrm{FF}[1,:] \leftarrow \boldsymbol{B}[1,:], \mathrm{FF}[:,1] \leftarrow \boldsymbol{B}[:,1]$
3: **for** $i,j \in \{2,..,N\}$ **do**
4:     $t_1 \leftarrow \mathrm{FF}[i-1,j]$
5:     $t_2 \leftarrow \mathrm{FF}[i,j-1]$
6:     $t_3 \leftarrow \mathrm{FF}[i-1,j-1]$
7:     $\mathrm{FF}[i,j] \leftarrow \boldsymbol{B}[i,j] \cdot (1 + \min_{j \in \{1,2,3\}} t_j)$
8: **Return** $\max_{i \in [N]} \mathrm{FF}[i,i]$

---

**A message-passing perspective of** $\mathrm{MSS}(\boldsymbol{B})$    Next, we conceptualize MSS as an instance of iterative message passing, akin to graph neural networks, applied to a directed acyclic graph where the nodes are initialized with binary states. This perspective enables us to develop a neural network capable of simulating Algorithm 1. Given the input matrix $\boldsymbol{B} = \boldsymbol{P}\boldsymbol{A}\boldsymbol{P}^\top$, we construct the graph $\mathcal{G}_N$ as a $N \times N$ directed acyclic grid graph. The nodes of $\mathcal{G}_N$ are $V(\mathcal{G}_N) = [N] \times [N]$, with each node $(i,j)$ having an in-neighbor set $\text{In-Nbr}((i,j)) := \{(i-1,j), (i,j-1), (i-1,j-1)\}$ except at the boundaries. The node embeddings of $\mathcal{G}_N$ are initialized as $h_0(i,j) = \boldsymbol{B}[i,j] \in \mathbb{R}$. At propagation layer $\ell \in [N]$, each node $(i,j)$ first receives messages from its in-neighbors $(s,t)$, then aggregates these messages, and finally updates its embedding as:

$$\mathcal{M}_l(i,j) = \cup_{(s,t) \in \text{In-Nbr}(i,j)} \mathrm{DPmsg}\big(h_\ell(s,t), h_\ell(i,j)\big)$$
$$h_{\ell+1}(i,j) = \mathrm{DPupdate}\big(h_0(i,j), \mathrm{DPaggr}(\mathcal{M}_l(i,j))\big). \tag{2}$$

Note that, unlike traditional GNNs, which use $h_\ell(i,j)$ as input at each layer, we feed the initial feature $h_0(i,j)$ as input at each propagation step, specifically tailoring our approach to simulate Algorithm 1. After the completion of $N$ of such propagations, we compute the output of the DP as

$$\mathrm{DPoutput}(\boldsymbol{B}) = \mathrm{DPreadout}\big(\cup_{(i,j) \in V(\mathcal{G}_N)} h_N(i,j)\big). \tag{3}$$

Note that when $\boldsymbol{B} \in \{0,1\}^{N \times N}$, the specific choices of the functions: $\mathrm{DPmsg}(x,y) = x$, $\mathrm{DPaggr}(\{x_i\}) = 1 + \min_i x_i$, $\mathrm{DPupdate}(x,y) = xy$, $\mathrm{DPreadout}(\{x\}) = \max\{x\}$ ensures that $\mathrm{DPoutput}(\boldsymbol{B}) = \mathrm{MSS}(\boldsymbol{B})$, as computed in Algorithm 1.

**Differentiable approximation of the combinatorial MCP in Eq.** (1)    Here, we propose a neural network $\omega_{\theta,\phi,\psi}$, with model parameters $\theta$, $\phi$, and $\psi$ (to be described later), to approximate the combinatorial MCP formulation (1). Our approach involves continuous relaxations of the original objective (1) on three fronts. **(1)** We relax the hard permutation matrix $\boldsymbol{P}$ into a soft permutation matrix $\boldsymbol{S}$, modeled as a doubly stochastic matrix. **(2)** It becomes difficult to attain zero values in the right entries of $\boldsymbol{S}\boldsymbol{A}\boldsymbol{S}^\top$ due to the continuous relaxation of $\boldsymbol{S}$. Moreover, the binary matrix $\boldsymbol{A}$ attenuates gradient signals. To address these problems we substitute the binary 0/1 values in the adjacency matrix $\boldsymbol{A}$ with continuous values, computed using node embeddings for each node $u \in V(G)$. **(3)** Since items (1) and (2) result in continuous relaxations of $\boldsymbol{P}\boldsymbol{A}\boldsymbol{P}^\top$, the initial states $h_0(i,j)$ now contain continuous values instead of binary 0/1 values. Consequently, we model the embeddings of the grid graph using continuous values.

— *Computation of node embeddings:* We use a graph neural network $\mathrm{GNN}_\theta$ with parameters $\theta$, to perform message passing across $L$ propagation layers to compute node embeddings $\boldsymbol{X} \in \mathbb{R}^{N \times d}$ where $\boldsymbol{X} = [\boldsymbol{x}_L(u) \in \mathbb{R}^d]_{u \in [N]}$. The embeddings $\boldsymbol{x}_L(u)$ capture the information about the subgraph containing nodes within $L$-hop distance from $u$. These embeddings are then used to perform continuous relaxations $\boldsymbol{W}$ of the adjacency matrix $\boldsymbol{A}$ as follows:

$$\boldsymbol{W} = \boldsymbol{A} \odot \boldsymbol{X}\boldsymbol{X}^\top, \quad \text{where,} \ \boldsymbol{X} = [\boldsymbol{x}_L(u) \in \mathbb{R}^d]_{u \in [N]} := \mathrm{GNN}_\theta(G) \tag{4}$$

— *Relaxation of hard to soft permutations:* The hard permutation matrix $\boldsymbol{P}$ in Eq. (1) is the essential blocker to an efficient and differentiable expression. To address this, we approximate $\boldsymbol{P}$ with a doubly stochastic soft permutation matrix $\boldsymbol{S}$. We obtain $\boldsymbol{S}$ by first feeding the node embeddings $\boldsymbol{X}$ into a neural network $T_\phi$, and then applying Sinkhorn iterations (Mena et al., 2018; Cuturi, 2013) to the resulting matrix $T_\phi(\boldsymbol{X})$. Given a maximum number of iterations $\tau_{\max}$, we compute $\boldsymbol{S}$ as:

$$\boldsymbol{S} = \boldsymbol{T}_{\tau_{\max}} \qquad \text{where, } \boldsymbol{T}_0 = \exp(T_\phi(\boldsymbol{X})/\lambda); \quad \boldsymbol{T}_{\tau+1} = \text{ColumnScale}(\text{RowScale}(\boldsymbol{T}_\tau)), \quad (5)$$

where $\lambda$ is a temperature parameter; RowScale and ColumnScale perform row and column normalizations of the matrix argument. As $\tau_{\max} \to \infty$, $\boldsymbol{S}$ approaches a doubly stochastic matrix and as $\lambda \to 0$ and $\tau_{\max} \to \infty$, $\boldsymbol{S}$ converges to a hard permutation matrix. Using Eqs. (4) and (5), we compute the relaxation of $\boldsymbol{P}\boldsymbol{A}\boldsymbol{P}^\top$ as: $\boldsymbol{B} = \boldsymbol{S}\boldsymbol{W}\boldsymbol{S}^\top$.

— *Relaxation of Algorithm 1:* Unlike the binary input in Algorithm 1, $\boldsymbol{B}$ now takes continuous values. Therefore, we use continuous $h_k(i,j) \in \mathbb{R}$ in Eqs. (2)– (3) and model DPmsg, DPupdate and DPreadout using neural networks (Details in Appendix D) with parameters $\psi$. Under these relaxations, we compute the node embeddings $h_\ell(i,j)$ on $\mathcal{G}_N$, starting with $h_0(i,j) = (\boldsymbol{S}\boldsymbol{W}\boldsymbol{S}^\top)[i,j]$ as follows:

$$h_{\ell+1}(i,j) = \text{DPupdate}_\psi \left( (\boldsymbol{S}\boldsymbol{W}\boldsymbol{S}^\top)[i,j], \textstyle\sum_{(s,t)\in\text{In-Nbr}(i,t)} \text{DPmsg}_\psi \big(h_\ell(i,j), h_\ell(s,t)\big) \right). \quad (6)$$

Finally, we predict the clique number $\omega(G)$ via max pooling, following Step 8 in Algorithm 1:

$$\omega_{\theta,\phi,\psi}(G) = \max_{(i,j)\in V(\mathcal{G}_N)} \text{DPreadout}_\psi(h_N(i,j)). \quad (7)$$

**Training** Given $I$ training instances $\{G_i, \omega(G_i) \,|\, i \in [I]\}$, we learn $\theta, \phi, \psi$ by minimizing the mean squared error $\mathcal{L}_{\text{MSS}} = \frac{1}{I} \sum_{i\in[I]} (\omega_{\theta,\phi,\psi}(G_i) - \omega(G_i))^2$.

**Limitation of MxNet (MSS)** As mentioned in Section 1.1, MxNet (MSS) can estimate $\omega(G)$ accurately, but, owing to massive symmetries in the distant supervision, succumbs to confusing gradient signals from a large number of equivalent soft permutations that achieve the optimal objective, and does not always succeed at producing interpretable "clique certificates" $\boldsymbol{S}$. The space of globally optimal $\boldsymbol{S}$s increases further because neither the combinatorial MCP (1) nor its neural approximation (7) constrains the position of the subsquare within the permuted matrix $\boldsymbol{B}$. We now proceed to rectify this limitation.

## 3.2 Design of MxNet (SubMatch)

**Finding cliques via subgraph matching** If the Gumbel-Sinkhorn network $T_\phi$ is powerful enough to find an effective soft permutation $\boldsymbol{S}$, it may well be powerful enough to arrange the maximal subsquare in a fixed position, say, $[1{:}\omega(G), 1{:}\omega(G)]$. How do we endow $T_\phi$ with this inductive bias?

We start with the standard observation that a clique $\mathcal{K}_c$ is a subgraph of $G$ (written as $\mathcal{K}_c \subseteq G$) if and only if $c \le \omega(G)$. If we had a perfect black box subgraph test, "$\mathcal{K}_2 \subseteq G$", "$\mathcal{K}_3 \subseteq G$", ..., "$\mathcal{K}_{\omega(G)} \subseteq G$" would pass, but "$\mathcal{K}_{\omega(G)+1} \subseteq G$" and subsequent tests would fail. Formally, $\mathcal{K}_1 \subset \mathcal{K}_2 \subset \ldots \mathcal{K}_{\omega(G)} \subseteq G$, but $\mathcal{K}_c \nsubseteq G$ for any $c > \omega(G)$.

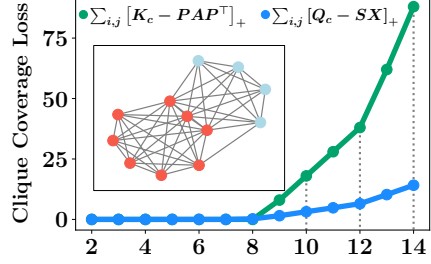

We shift our perspective on $\boldsymbol{S}$ as "a permutation that collects clique nodes into a contiguous subsquare" to "a permutation that maps $\mathcal{K}_c$ injectively into $G$", where each $\mathcal{K}_c$ is presented in a specific format. $\boldsymbol{K}_c$, the adjacency matrix of $\mathcal{K}_c$, is zero-padded to an $N \times N$ adjacency matrix, but the edges are always presented in the $[1{:}c, 1{:}c]$

Figure 2: Testing if $\mathcal{K}_c \xrightarrow{c} \subseteq G$ for increasing $c$ until $c > \omega(G){=}8$.

upper left corner. Formally, $\boldsymbol{K}_c = \begin{bmatrix} \boldsymbol{1}_{c\times c} & \boldsymbol{0}_{c\times(N-c)} \\ \boldsymbol{0}_{(N-c)\times c} & \boldsymbol{0}_{(N-c)\times(N-c)} \end{bmatrix}$. Then there exists a hard node permutation $\boldsymbol{P}$ such that

$$\boldsymbol{K}_1 \le \boldsymbol{K}_2 \le \cdots \le \boldsymbol{K}_{\omega(G)} \le \boldsymbol{P}\boldsymbol{A}\boldsymbol{P}^\top; \quad \boldsymbol{K}_c \nleq \boldsymbol{P}\boldsymbol{A}\boldsymbol{P}^\top \; \forall c > \omega(G) \text{ (elementwise inequality). (8)}$$

The inequality $\boldsymbol{K}_c < \boldsymbol{P}\boldsymbol{A}\boldsymbol{P}^\top$ can be written equivalently as $[\boldsymbol{K}_c - \boldsymbol{P}\boldsymbol{A}\boldsymbol{P}^\top]_+ = 0$. The asymmetric distance $\big\|[\boldsymbol{K}_c - \boldsymbol{P}\boldsymbol{A}\boldsymbol{P}^\top]_+\big\|_{1,1}$ measures the coverage of $\mathcal{K}_c$ by the graph $G$ and can be used as distance measure for subgraph matching (Lou et al., 2020; Ranjan et al., 2022; Roy et al., 2022). Then, using the nested inequality in Eq. (8), we will have $[\boldsymbol{K}_c - \boldsymbol{P}\boldsymbol{A}\boldsymbol{P}^\top]_+ = \boldsymbol{0}$ for all $c \le \omega(G)$ and for $c > \omega(G)$, we will have $[\boldsymbol{K}_c - \boldsymbol{P}\boldsymbol{A}\boldsymbol{P}^\top]_+ > \boldsymbol{0}$. The green line in Figure 2 shows

$\left\|[\boldsymbol{K}_c - \boldsymbol{P}\boldsymbol{A}\boldsymbol{P}^\top]_+\right\|_{1,1}$ as $c$ increases. As expected, with the ideal $\boldsymbol{P}$, it jumps from zero to positive values for $c > \omega(G) = 8$.

**Neural relaxation**    We employ $\text{GNN}_\theta$, introduced in Eq. (4), to represent graphs $G$ and $\mathcal{K}_c$ as $N \times d$ contextual node embedding matrices $\boldsymbol{X}$ and $\boldsymbol{Q}_c$. We replace the hard clique coverage loss $\sum_{i,j}[\boldsymbol{K}_c - \boldsymbol{P}\boldsymbol{A}\boldsymbol{P}^\top]_+[i,j]$ with a soft clique coverage loss $\rho_{\theta,\phi}(c;G) = \sum_{i,j}[\boldsymbol{Q}_c - \boldsymbol{S}\boldsymbol{X}]_+[i,j]$, where $\boldsymbol{S}$ is the soft permutation matrix (5). Consistent with $\boldsymbol{K}_c$, the first $c$ rows of $\boldsymbol{Q}$ contain the non-trivial embedding vectors corresponding to clique nodes $V(\mathcal{K}_c)$, whereas the lower $N - c$ rows are zero pads. The blue line in Figure 2 shows $\rho_{\theta,\phi}(c;G)$ against $c$, after $\theta, \phi$ are reasonably trained. We can again see a detectable jump after $c = \omega(G)$.

**Training curriculum**    Instead of using MXNET (MSS) to directly estimate $\omega(G)$, we will train $\rho_{\theta,\phi}(c;G)$ using a 'curriculum' that encourages the neural counterpart of Eqn. (8) to hold. Specifically, we seek to minimize $\min_{\theta,\phi} \mathcal{L}_{\text{SubMatch}}(\omega(G); \{\rho_{\theta,\phi}(c;G)\}_{c \geq 2})$ where $\mathcal{L}_{\text{SubMatch}}$ is written as

$$\mathcal{L}_{\text{SubMatch}} = \sum_{i \in [I]} \Bigg( \sum_{c=2}^{\omega(G)} \frac{\rho_{\theta,\phi}(c;G)}{|\omega(G) - 1|} + \sum_{c=2}^{\omega(G)-1} \frac{[\rho_{\theta,\phi}(c+1;G) - \rho_{\theta,\phi}(c;G) - \gamma]_+}{|\omega(G) - 2|} \\ + [\rho_{\theta,\phi}(\omega(G);G) - \rho_{\theta,\phi}(\omega(G)+1;G) + \delta]_+ \Bigg) \quad (9)$$

**(I)**  The first term encourages $\boldsymbol{S}$ to discover that $\mathcal{K}_c \subseteq G$ for $c \in [2, \omega(G)]$.
**(II)**  As long as $c < \omega(G)$, we do not want $\rho_{\theta,\phi}(c;G) \gg \rho_{\theta,\phi}(c+1;G)$. The second term constitutes this "nesting curriculum".
**(III)**  The third term encourages the jump: $\rho_{\theta,\phi}(\omega(G)+1;G) \gg \rho_{\theta,\phi}(\omega(G);G)$.

Here $\delta$ and $\gamma$ are margin hyperparameters, with $\gamma \ll \delta$ typically giving the best performance. In Eq. (9), we may limit the inner summation of the first term to only $c = \omega(G)$ and the inner summation of the last term to $c = \omega(G) - 1$. However, summation from 2 to $\omega(G)$ provides more explicit guidance to the learner that, $K_2, \ldots, K_c$ are subgraphs of $G$. Our experiments in Appendix F show that it is unable to reason that if $K_c$ is a subgraph of $G$ then also $K_2, \ldots$ are also subgraphs of $G$.

**Inference**    During inference, we evaluate $\rho_{\theta,\phi}(c;G)$ as we increment $c$, reporting our estimate $\widehat{\omega}(G)$ as the smallest $c^*$ such that $\rho_{\theta,\phi}(c^*+1;G) \geq \rho_{\theta,\phi}(c^*;G) + \delta$.

**Pros and cons**    Loss expression (9) achieves two goals. It guides the search for $\boldsymbol{S}$ using a progression from smaller to largest cliques, and it guides the positioning of the MSS to the upper left corner for a variety of clique sizes. Together, these improve the interpretability of "clique certificates" (somewhat dense subsquares in the continuous regime). Experiments suggest that MXNET (MSS) can sometimes beat MXNET (SubMatch) in terms of the raw accuracy of estimating $\omega(G)$.

### 3.3    Design of MXNET (Composite)

Given their complementary strengths, we are naturally motivated to combine MXNET (MSS) and MXNET (SubMatch) into our final proposal, MXNET (Composite).

**Training**    We simultaneously train the neural networks $\omega_{\theta,\phi,\psi}$ and $\rho_{\theta,\phi}(c;G)$ by minimizing with a composite loss function, with the mixing hyperparameter $\lambda$, written as

$$\mathcal{L}_{\text{Composite}} = \lambda \mathcal{L}_{\text{MSS}} + \mathcal{L}_{\text{SubMatch}} \quad (10)$$

**Bicriteria early stopping**    Early experiments suggested that the two parts of the composite loss can be oppositional. Therefore, we developed a bicriteria early stopping logic. It consists of two phases. Initially, we monitor the reduction in $\mathcal{L}_{\text{SubMatch}}$, indicating the model's effectiveness in identifying an optimal permutation that satisfies the query clique detection challenges. Once $\mathcal{L}_{\text{SubMatch}}$ saturates, we transition to monitoring $\mathcal{L}_{\text{MSS}}$. To prevent the second phase from reversing the gains of the first, we allow/accept a reduction in $\mathcal{L}_{\text{MSS}}$ only if the damage to $\mathcal{L}_{\text{SubMatch}}$ is below a threshold. This enhanced early stopping policy ensures a balanced optimization of (10).

**Inference**    During inference, we have the option to base our predictions on either $\omega_{\theta,\phi,\psi}$ or $\rho_{\theta,\phi}(c;G)$. While $\rho_{\theta,\phi}(c;G)$ provides a certificate for the maximum detected clique, $\omega_{\theta,\phi,\psi}$ directly predicts the clique number based on neural maximum subsquare detection (7). Given our objective of accurate clique number prediction, we choose to use the prediction of $\omega_{\theta,\phi,\psi}$.

## 4    Experiments

We report on extensive experiments using eight datasets, comparing the performance of MXNET with other methods. We also instrument different components of MXNET to understand their impact.

### 4.1 SETUP

**Datasets** We conduct experiments on eight datasets, comprising five real-world and three synthetic datasets. Real-world datasets include (1) IMDB-BINARY (IMDB), (2) Enzymes and modular products of graph pairs from (3) PTC-MM-$m$, (4) AIDS, (5) Mutagenicity (MUTAG-$m$) datasets. We also generate three synthetic datasets from (6) DSJC, (7) Brockington (Brock), and (8) RB. One key application of clique number estimation is to compute the similarity score between two graphs based on the size of their maximum common induced subgraph, expressed as the modular product[*] between the graph pairs. Motivated by applications in graph similarity, we use modular graph products for three datasets, *viz.*, AIDS, MUTAG, PTC-MM. We call them AIDS-$m$, MUTAG-$m$ and PTC-MM-$m$ respectively. Additional details are in in Appendix E.

**Baseline neural and neural+combinatorial MCP solvers** Prior methods that operate under the three practically motivated métiers of our problem setting — end-to-end differentiability, a focus on clique number prediction rather than a maximal clique itself, and the ability to train under distant supervision — are surprisingly hard to find. Plausible baselines from the neural network and machine learning community include six unsupervised methods, *viz.*, (1) EGN (Karalias & Loukas, 2020), (2) SCT (Min et al., 2022), (3) SFE (Karalias et al., 2022), (4) NSFE (Karalias et al., 2022), (5) ST (Bengio et al., 2013), (6) Reinforce (Williams, 1992); two variants of Difusco (Sun & Yang, 2024), which are trained under extreme supervision of explicit clique demonstration, *viz.*, (7) Difusco (Cont) and (8) Difusco (Cat), which use continuous (Gaussian) and categorical sampling at the de-noising stage; one neural subgraph counting based approach trained under distant supervision– (9) NeurSC (Wang et al., 2022); and two models trained under Reinforcement Learning, *viz.*, (10) GFnet (Zhang et al., 2024) and (11) VAG-CO (Sanokowski et al., 2023). None of these methods is trained under distant supervision, nor are they able to estimate the clique numbers without making implicit or explicit hard decisions about which nodes constitute the clique.

**Heavy-lifting decoders** A differentiable network, as well as many heuristic MCP solvers or relaxations of integer programs, will usually associate a score with each node $u \in V(G)$ that $u$ indicating its membership in the maximal clique it has identified. We call this a **heatmap** on the nodes. To demonstrate a concrete clique, these methods then feed these heatmap into the **decoder**, which is generally *procedural*, not declarative, and therefore not differentiable. EGN, SCT, Difusco, GFnet, and VAG-COuse such non-differentiable decoders. Appendix E.3 discusses decoders at length.

**Evaluation** In the face of such heterogeneity and "culture differences", it is difficult to compare competing systems with a single yardstick. We first present comparisons segregated between methods with and without decoders. MxNET easily beats all non-decoder methods, and even most decoder-based methods on many data sets. Later, we allow decoders, but we must carefully control the computational costs of sampling inside decoders. Some more details are in Appendix E.3.

We split each dataset $D = \{G_i, \omega(G_i) \,|\, i \in [I]\}$ into 60% training, 20% eval, and 20% test folds. We report Mean Squared Error (MSE) between the predicted clique number and the ground truth clique number. Appendix E.4 explains why traditional metrics like approximation ratio are not suitable.

### 4.2 RESULTS

**Comparison with baselines** Here, we compare the performance of MxNET (Composite) against all the ten neural baselines in terms of MSE. We use two variants of MxNET, *viz.*, MxNET (**ES:MSS**) which uses early stopping using MSS loss $\mathcal{L}_{\text{MSS}}$ and MxNET (**ES:Bi**) which uses bi-criteria early stopping (Section 3.3). Table 1 summarizes the results. The key observations are as follows. **(1)** Although six out of ten baselines use combinatorial decoders during inference, and two of them use extreme supervision, MxNET achieves the best performance on five datasets and the second best performance on two datasets. In these datasets, EGN is the second-best performer in the first five and the best in the last three. **(2)** The baselines that do not use a decoder (the last four methods in Table 1) are massively outperformed by MxNET. In fact, they are outperformed by every baseline that uses a decoder. Remarkably, MxNET is the only model that does not use a decoder and yet not only competes with but often surpasses the decoder-based baselines. **(3)** Overall, GFnet is the third-best performer in four out of eight datasets. **(4)** Difusco (Cat) consistently outperforms Difusco (Cont) across all datasets, aligning with the observations reported in their original paper.

**Ablation study on variants of MxNET.** We compare the performance of the three variants of our method, *viz.*, MxNET (MSS), MxNET (SubMatch) and MxNET (Composite), whose training

---

[*]https://en.wikipedia.org/wiki/Modular_product_of_graphs

Table 1: Performance in terms of mean squared error (MSE) between predicted and true clique numbers for two variants of MxNet (MxNet (**ES:MSS**), MxNet (**ES:Bi**)), EGN (Karalias & Loukas, 2020), SCT (Min et al., 2022), Difusco (Cont, Cat) (Sun & Yang, 2024), GFnet (Zhang et al., 2024), VAG-CO (Sanokowski et al., 2023), SFE (Karalias et al., 2022), NSFE (Karalias et al., 2022), ST (Bengio et al., 2013), Reinforce (Williams, 1992) and NeurSC (Wang et al., 2022) on 20% test set. MxNet (**ES:MSS**) uses early stopping using $\mathcal{L}_{MSS}$, whereas MxNet (**ES:Bi**) uses bicriteria early stopping described in Section 3.3. In all cases, we use MxNet (Composite), which is trained using the composite loss (10). First six (last seven) methods (do not) use decoder. Green, blue, and yellow show the best, second-best, and third-best methods, respectively.

| | | IMDB | AIDS-$m$ | PTC-MM-$m$ | DSJC | Brock | Enzymes | RB | MUTAG-$m$ |
|---|---|---|---|---|---|---|---|---|---|
| **Decoder** | EGN (Karalias & Loukas, 2020) | 0.102 | 0.610 | 0.284 | 0.030 | 1.310 | 0.109 | 15.615 | 1.010 |
| | SCT (Min et al., 2022) | 4.102 | 3.865 | 1.802 | 92.675 | 35.885 | 0.891 | 50.230 | 11.105 |
| | Difusco (Cont) (Sun & Yang, 2024) | 5.333 | 4.665 | 2.621 | 59.820 | 39.040 | 1.034 | 63.490 | 10.590 |
| | Difusco (Cat) (Sun & Yang, 2024) | 1.361 | 4.170 | 2.244 | 1.560 | 10.595 | 0.950 | 55.630 | 10.965 |
| | GFnet (Zhang et al., 2024), | 2.815 | 0.740 | 0.554 | 4.820 | 8.975 | 0.857 | 34.200 | 2.565 |
| | VAG-CO (Sanokowski et al., 2023) | 5.037 | 33.905 | 19.962 | 7.665 | 15.450 | 4.244 | 320.240 | 132.700 |
| **Non-Decoder** | SFE (Karalias et al., 2022) | 4.833 | 44.340 | 26.568 | 12.465 | 53.595 | 1.630 | 72.020 | 110.905 |
| | NSFE (Karalias et al., 2022) | 3.185 | 36.005 | 19.800 | 5.255 | 47.920 | 1.118 | 78.890 | 96.445 |
| | ST (Bengio et al., 2013) | 8.972 | 44.885 | 26.703 | 5.410 | 55.740 | 5.454 | 131.230 | 111.020 |
| | Reinforce (Williams, 1992) | 16.472 | 59.170 | 37.455 | 13.280 | 403.095 | 1.731 | 229.435 | 132.700 |
| | NeurSC (Wang et al., 2022) | 5.330 | 1.240 | 1.510 | 15.350 | 88.790 | 0.630 | 10.080 | 3.520 |
| | MxNet (**ES:MSS**) | 0.094 | 0.341 | 0.179 | 0.383 | 10.022 | 0.203 | 6.875 | 4.130 |
| | MxNet (**ES:Bi**) | 0.094 | 0.341 | 0.179 | 0.383 | 10.022 | 0.196 | 6.875 | 0.613 |

Table 2: Comparison of MxNet (MSS), MxNet (SubMatch), and MxNet (Composite) in terms of Mean Squared Error (MSE) on eight datasets. Numbers in green, blue indicate the best performers and second best performers respectively.

| | IMDB | AIDS-$m$ | PTC-MM-$m$ | DSJC | Brock | Enzymes | RB | MUTAG-$m$ |
|---|---|---|---|---|---|---|---|---|
| MxNet (MSS) | 24.434 | 0.295 | 2.364 | 66.890 | 150.430 | 0.205 | 10.190 | 4.130 |
| MxNet (SubMatch) | 0.056 | 0.420 | 0.512 | 2.820 | 150.890 | 0.210 | 7.170 | 4.040 |
| MxNet (Composite) | 0.094 | 0.341 | 0.179 | 0.383 | 10.022 | 0.196 | 6.875 | 0.613 |

losses are $\mathcal{L}_{MSS}$, $\mathcal{L}_{SubMatch}$ (9) and $\mathcal{L}_{Composite}$ (10), respectively. Table 2 shows the results. The key observations are as follows. **(1)** MxNet (Composite) emerges as the exclusively best performer in six out of eight datasets, with a comparable performance to MxNet (MSS) in Enzymes, and being outperformed by MxNet (MSS) in DSJC. Particularly noteworthy is the substantial improvement in MxNet's performance in datasets like IMDB, MUTAG-$m$, and RB, highlighting the efficacy of combining $\mathcal{L}_{MSS}$ and $\mathcal{L}_{SubMatch}$ for accurate clique prediction tasks; **(2)** In five out of eight datasets, MxNet (MSS) significantly outperforms MxNet (SubMatch), thereby demonstrating its better capability at predicting the clique number directly. **(3)** MxNet (SubMatch) is never the top performer on its own, highlighting that while it provides important signals for MxNet's performance, its indirect signal about clique sizes is insufficient for accurately predicting clique number.

Table 3: Performance of MxNet (Composite) for different early stopping criteria, which are based on $\mathcal{L}_{MSS}$, $\mathcal{L}_{SubMatch}$ and $\mathcal{L}_{Composite}$ respectively. For each of these early stopping criteria, we use MxNet (MSS) and MxNet (SubMatch) (denoted as MSS and SubMatch) components to measure the performance in terms of MSE on the test set. Numbers in green indicate the best performers.

| | IMDB | | AIDS-$m$ | | PTC-MM-$m$ | | DSJC | |
|---|---|---|---|---|---|---|---|---|
| | MSS | SubMatch | MSS | SubMatch | MSS | SubMatch | MSS | SubMatch |
| MxNet (**ES:MSS**) | 0.094 | 0.241 | 0.341 | 0.705 | 0.179 | 0.265 | 0.383 | 9.360 |
| MxNet (**ES:SubMatch**) | 0.331 | 0.065 | 0.307 | 0.450 | 0.232 | 0.495 | 0.581 | 7.335 |
| MxNet (**ES:Bi**) | 0.094 | 0.241 | 0.341 | 0.705 | 0.179 | 0.265 | 0.383 | 9.360 |

| | Brock | | Enzymes | | RB | | MUTAG-$m$ | |
|---|---|---|---|---|---|---|---|---|
| | MSS | SubMatch | MSS | SubMatch | MSS | SubMatch | MSS | SubMatch |
| MxNet (**ES:MSS**) | 10.022 | 726.995 | 0.203 | 0.353 | 6.875 | 35.620 | 4.130 | 7.805 |
| MxNet (**ES:SubMatch**) | 19.165 | 170.280 | 0.196 | 0.210 | 10.167 | 6.830 | 4.151 | 4.040 |
| MxNet (**ES:Bi**) | 10.022 | 726.995 | 0.196 | 0.210 | 6.875 | 35.620 | 0.613 | 4.680 |

**Effect of bicriteria early stopping** We investigate the impact of training MxNet (Composite) under the bicriteria early stopping mechanism (MxNet (**ES:Bi**)), by comparing it against two early stopping criteria, based solely on $\mathcal{L}_{MSS}$ (MxNet (**ES:MSS**)) and $\mathcal{L}_{SubMatch}$ (MxNet (**ES:SubMatch**)). We use the inference methods corresponding to both MxNet (MSS) and MxNet (SubMatch) on the test set. Table 3 reports the MSE numbers, which reveal the following observations. **(1)** MxNet (**ES:MSS**) shows extremely strong performance across all datasets, consistently achieving the best MSE results, when the inference is performed using MxNet (MSS). However, as often seen, most notably in MUTAG-$m$, RB and Brock, this sometimes comes at a

significant cost of MSE resulted from inference using MxNet (SubMatch), which indicates potential overfitting. **(2)** MxNet (`ES:SubMatch`) achieves best MSE results across five out of eight datasets, when inference is performed using MxNet (SubMatch). However the performance measured in terms of MSE resulted from MxNet (MSS) based inference is not as strong, except for Enzymes where it matches the best performance; **(3)** MxNet (`ES:Bi`) provides a more balanced approach, and we found bicriteria early stopping to be an effective guardrail against potential overfitting of MxNet (MSS). As seen in Table 3, MxNet (`ES:Bi`) prevents extreme overfitting in datasets like MUTAG-$m$, RB and Brock, while matching the MSE achieved by MxNet (MSS) in datasets such as IMDB, AIDS-$m$, PTC-MM-$m$, and Enzymes. Therefore, while MxNet (`ES:Bi`) does not directly enhance performance, it ensures that inference via MxNet (SubMatch) does not degrade significantly. This makes bicriteria early stopping a reliable mediator.

**Contribution of combinatorial decoder** Here, we compare the performance of each decoder based neural model against its corresponding non-neural variant (named as Heuristic), which consists of only *its own constituent decoder*. The initial node heatmap input to the decoder is computed using a structure guided node score, selected from node degree, PageRank and clustering coefficient, based on minimum validation error. In addition, we also empower all methods by increasing the number of clique proposals by 4x from Table 1. Table 4 presents the results on two datasets in terms of both MSE and time. We observe that: **(1)** In some cases, the non-neural Heuristic variants are comparable to or even outperform the neural variants. This is particularly noticeable in SCT baseline for all variants, Difusco (Cat) in PTC-MM-$m$, and GFnet in IMDB. This highlights the significant role

Table 4: Decoder vs No-decoder. green, blue, and yellow indicate the best, second-best, and third-best performers

|  | IMDB | | PTC-MM-$m$ | |
|---|---|---|---|---|
|  | MSE | Time (s) | MSE | Time (s) |
| EGN | 0.102 | 0.117 | 0.284 | 0.149 |
| EGN (4x) | 0.102 | 0.236 | 0.183 | 0.410 |
| Heuristic | 2.824 | 0.146 | 0.861 | 0.159 |
| Heuristic(4x) | 0.222 | 0.137 | 0.497 | 0.493 |
| SCT | 4.102 | 0.051 | 1.802 | 0.039 |
| SCT (4x) | 0.259 | 0.048 | 0.701 | 0.063 |
| Heuristic | 3.287 | 0.002 | 0.922 | 0.016 |
| Heuristic (4x) | 0.250 | 0.007 | 0.480 | 0.030 |
| Difusco (Cat) | 1.361 | 0.789 | 2.244 | 0.690 |
| Difusco (Cat) (4x) | 0.231 | 0.722 | 0.979 | 0.799 |
| Heuristic | 2.741 | 0.011 | 0.931 | 0.045 |
| Heuristic (4x) | 2.741 | 0.010 | 0.931 | 0.058 |
| GFnet | 2.815 | 0.032 | 0.554 | 0.011 |
| GFnet (4x) | 0.167 | 0.081 | 0.166 | 0.035 |
| Heuristic | 2.333 | 0.018 | 2.623 | 0.027 |
| Heuristic (4x) | 0.213 | 0.053 | 1.160 | 0.143 |
| MxNet | 0.094 | 0.029 | 0.179 | 0.018 |

of the decoder in the clique detection tasks. **(2)** While MxNet still outperforms all baselines in IMDB despite their improvements, it ranks third in PTC-MM-$m$, behind EGN (4x) and GFnet (4x). However, this improvement in the baselines comes with an almost 2x increase in run-time, making our method more efficient in terms of inference time latency.

**Effect of distribution shift** We construct an OOD (out-of-distribution) dataset based on the original PTC-MM-$m$ dataset as follows: Given the dataset $D = (G_i, \omega(G_i))|_{i=1}^{|I|}$, we select all graphs with $|V_i| \geq 80$ and $\omega_i \geq 7$ for the test set, while randomly partitioning the remaining graphs into training and validation splits. Consequently, during training, neither MxNet nor the baselines are exposed to large graphs with large clique sizes. We then evaluate the performance of all methods in predicting clique numbers on the test set. As the test dataset is out-of-distribution, we observe performance drops for all methods (Table 5). Nevertheless, MxNet consistently outperforms others.

Table 5: Performance in terms of mean squared error (MSE) between the predicted and true clique numbers for two variants of PTC-MM-$m$ dataset: (1) Default: which has been used in all experiments, and (2) OOD: which consists of a specialized test set with all graphs containing $|V| \geq 80$ and $\omega(G) \geq 7$. Numbers in green, blue indicate the best and second best performers respectively.

|  | EGN | SCAT | GFNET | SFE | NSFE | ST | REINFORCE | MXNET |
|---|---|---|---|---|---|---|---|---|
| PTC-MM-$m$ (Default) | 0.284 | 1.802 | 0.554 | 26.568 | 19.800 | 26.703 | 37.455 | 0.179 |
| PTC-MM-$m$ (OOD) | 3.26 | 1.256 | 0.832 | 42.328 | 33.129 | 42.644 | 56.682 | 0.822 |

## 5 Conclusions

The proposed MCP method, MxNet, introduces an end-to-end differentiable neural approach for predicting clique numbers of graphs. It can seamlessly integrate into existing deep learning frameworks (for retrieval applications, *e.g.*), where clique numbers serve as critical latent signals. By combining MxNet (MSS) and MxNet (SubMatch), we achieve a robust balance between high prediction accuracy and interpretability. Future work could focus on enhancing interpretability, handling dynamic graphs, and integrating domain knowledge through dense node/edge features within relevance learning frameworks. This research also paves the way for exploring differentiable predictions of other combinatorial optimization challenges on graphs, extending applications in graph analysis and beyond.

ACKNOWLEDGEMENTS

IR acknowledges Google PhD Fellowship. SC acknowledges IBM Grant and SERB grant. AD acknowledges SERB CRG grant.

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

# Clique Number Estimation via Differentiable Functions of Adjacency Matrix Permutations
## (Appendix)

CONTENTS

## A    LIMITATIONS

While MXNET presents an end-to-end differentiable mechanism for robust prediction of clique numbers in graphs, several potential limitations should be considered.

1. The method's reliance on node permutations and subgraph matching may lead to memory or computational constraints when dealing with large or dense graphs. Although scaling concerns are common among combinatorial and neural approaches, our method still warrants additional attention in this regard.

2. In various graph applications where cliques are important latent signals, graphs often come with domain knowledge encoded as dense node or edge features. While our integration with common GNN-based encoders provides a solution to incorporate such domain information, disentangled modeling of these influences in relevance judgments may be needed.

3. While MXNET demonstrates remarkable capability in navigating the exponentially large space of permutations, as seen in the experiments, there is still room for improvement, especially compared to combinatorial methods or decoder-based pipelines. This requires further investigation into improving model decisions, perhaps through better search techniques leading to more interpretable certificates of clique presence.

4. This proposal highlights significant opportunities for developing fully differentiable and backpropagable neural pipelines for solving combinatorial problems on graphs. While existing neural architectures have successfully reduced amortized solution costs compared to combinatorial approaches, these methods could not be seamlessly integrated into existing neural network pipelines. While our proposal addresses clique number estimation tasks, extension to other combinatorial challenges requires further research.

## B    BROADER IMPACT

Our proposed work on MXNET has significant broader impact across various domains of research and applications.

1. In biological networks, cliques represent highly connected functional units such as protein complexes and regulatory modules (Spirin & Mirny, 2003). Our neural clique number prediction module can assist in studying these biological systems and their functions.

2. In social networks, cliques represent tightly interconnected communities. Our model can provide insights related to community detection, influence propagation, maximization, and studying other network dynamics.

3. In recommendation systems, cliques often represent closely connected sets of items and users. Additionally, structural notions of pairwise similarity, such as the maximum common induced subgraph (MCIS), use clique number prediction as a subroutine. Accurate prediction of clique numbers can enhance recommendation pipelines by identifying relevant items and modeling user interest overlap.

4. Furthermore, the combination of MXNET (MSS) and MXNET (SubMatch) ensures not only high prediction accuracy but also interpretability. This is crucial in applications where understanding the reasoning behind predictions is essential for decision-making.

5. Since our proposed method MXNET primarily focuses on improving the accuracy and interpretability of predicting clique numbers in graphs using neural networks, there are no significant ethical concerns. The method does not involve human subjects or personal data, and it is not directly applicable to negative societal impacts.

## C   FURTHER DISCUSSION OF RELATED WORK

**Combinatorial methods for maximum clique detection**   The *maximum clique problem (MCP)* is NP-complete (Karp, 2010). It is notoriously hard to even approximate (Feige et al., 1991; Arora et al., 1998). Strong motivation from applications have nevertheless driven the development of combinatorial and math programming methods to find large cliques in real-life scenarios. The Bron–Kerbosch algorithm (Bron & Kerbosch, 1973) is among the well-known enumerative algorithms for MCP. Other systems (Choudhary et al., 2015) are tailored for social network analysis in massive graphs. Lu et al. (2017); Chang (2019) provide heuristics for exact maximum clique detection. He et al. (2023a) is for approximate k-clique detection with theoretical guarantee. One can adapt this technique to predict the clique number.

**Neural models for maximum clique detection**   Neural maximum clique detection methods fall into three main categories: supervised learning, unsupervised learning, and reinforcement learning.

— *Supervised approaches:* Recent advancements, such as Difusco (Sun & Yang, 2024), propose a diffusion-based generative model trained using node-level supervision of ground truth cliques, enabling non-autoregressive one-shot prediction of cliques. However, given the NP-Completeness of the problem, obtaining large quantities of labeled data with fine-grained optimal node-labels is impractical, which is essential for providing explicit guidance to training algorithms regarding clique structures.

— *Unsupervised approaches:* Consequently, the research community has shifted towards unsupervised (Karalias et al., 2022; Karalias & Loukas, 2020; Min et al., 2022) or reinforcement learning (Sanokowski et al., 2023; Zhang et al., 2024) approaches. Unsupervised methods learn a parameterized distribution over candidate solutions (Karalias & Loukas, 2020; Min et al., 2022), generating node-level heatmaps that indicate the likelihood of node participation in cliques. Yet, due to the problem's multimodal nature, such approaches resort to a greedy decoder with built-in clique maximization heuristics for autoregressive clique generation, limiting scalability and rendering end-to-end differentiable pipelines unfeasible. Our experiments reveal that the performance of such approaches heavily relies on the non-differentiable greedy decoder, as its removal leads to considerable performance degradation. Interestingly, these methods underscore the potency of decoding heuristics, where synergies between random graph structure-based signals and the decoder yield comparable or superior performance to neural mechanisms.

— *Reinforcement learning approaches:* In a concurrent line of research, reinforcement learning models (Sanokowski et al., 2023; Zhang et al., 2024) frame maximum clique detection as a sequential decoding process, training a Markov Decision Process (MDP) with a final reward based on detected maximum clique size. While such methods offer high accuracy and reduced amortized solution generation times, the costly decoding process and sparse reward structure render them incompatible with certain applications. Additionally, our experiments indicate that substituting the learned policy with graph statistic-based heuristics often yields comparable performance.

**Neural Algorithmic Reasoning (NAR)**   In recent years, a body of work study the design of neural network to mimic procedural algorithms (*e.g.*, sorting, searching, fundamental graph algorithms, *etc*), predominantly by leveraging the message passing module of graph neural network (GNN) (Veličković & Blundell, 2021; Bevilacqua et al., 2023; Rodionov & Prokhorenkova, 2024; Georgiev et al., 2024; Xu & Veličković, 2024; Ibarz et al., 2022; Veličković et al., 2022; Jürß et al., 2024; Bohde et al., 2024). Veličković & Blundell (2021) introduced this problem, where they provided intermediate states of the underlying procedural algorithm into the underlying GNN. Bevilacqua et al. (2023) apply causal regularization on top of the NAR framework. Rodionov & Prokhorenkova (2024) tackles NAR with intermediate states of the underlying algorithm. Veličković et al. (2022) performs benchmarking on a wide variety of popular algorithms from CLRS textbook. Very recently, Bohde et al. (2024) designs NAR model which preserves Markov property of algorithmic execution.

**Related graph search problems**   The database community has witnessed much recent interest in the closely related problem of counting the number of occurrences of a query graph in a corpus graph. NeurSC (Wang et al., 2022) starts by finding, for each node $u$ in query graph, plausible corpus graph nodes $v$ to which it may map. While doing this, NeurSC takes advantage of labels of $u, v$ and their neighborhoods. The query and each candidate graph in turn are then encoded via GNNs in both early (inter-graph) and late (intra-graph) interaction modes, and then these graph representations compared using an optimal transport method to arrive at a (soft) boolean prediction of whether the

query graph is a subgraph of the candidate. SCOPE (Li & Yu, 2024a) provides a combinatorial algorithm for subgraph counting based on tree decomposition of the pattern (similar to a query) graph, then using specific counting techniques (tISO-match) for trees and their combinations. These counting methods can be used for clique number prediction. We can invoke one of these methods to count the number of occurrences of $\mathcal{K}_c$ in the corpus graph for $c = 1, 2, \ldots$, after which $\omega(G)$ is predicted as the largest $c$ for which a positive count is reported. Given a subgraph $S$ of graph $G$, the density of $S$ is the ratio of the number of edges to the number of nodes in $S$. The *densest subgraph problem* is that of finding a subgraph of maximum density. Yet another problem is to find the *$k$-clique densest subgraph* (He et al., 2023a; Tsourakakis et al., 2013). Here, given an input graph, the goal is to detect a subgraph that maximizes the ratio between the number of $k$-cliques contained in it and the number of vertices in it. The densest subgraph problem is a special case with $k = 2$.

# D  IMPLEMENTATION DETAILS

## D.1  DETAILS OF OUR NETWORK ARCHITECTURE

Our approach consists of three key components: the graph neural network $\text{GNN}_\theta$, transformation network $T_\phi$, and neural relaxations $\text{DPmsg}_\psi$, $\text{DPupdate}_\psi$, and $\text{DPreadout}_\psi$. The graph neural network $\text{GNN}_\theta$ consists of a $L = 5$ layer graph neural network, that projects the initial node features to a $d = 10$ dimension embedding space. For all datasets, we set $\boldsymbol{x}_u(0) = [1]$ as the initial node features. Following the methodology proposed in (Li et al., 2015; 2019), we compute the updated node embeddings as follows using a Gated Recurrent Unit (GRU) as follows:
$\boldsymbol{x}_\ell(u) = \text{GRU}_\theta\left(\boldsymbol{x}_{\ell-1}(u), \sum_{v \in \text{nbr}(u)} \text{LINEAR}_\theta(\boldsymbol{x}_{\ell-1}(u), \boldsymbol{x}_{\ell-1}(v))\right)$ where $\text{LINEAR}_\theta$ is a single linear layer with both input dimension and output dimensions set to 10, and $\text{GRU}_\theta$ is the GRU network where the computed node embedding is utilized in the hidden state. After $L = 5$ such iterations, we derive $\boldsymbol{X} = [\boldsymbol{x}_L(u)]_{u \in [N]}$ as the final node embeddings. The transformation network $T_\phi$ consists of a 2-layer feedforward network which consists of a linear layer followed by ReLU activation and a linear layer. This maps the embeddings $\boldsymbol{X}$ from $d = 10$ dimensions to $N$ dimensions, where $N$ represents the number of nodes in the padded graph. Following the Sinkhorn iterations with $\tau_{\max} = 20$, where the Sinkhorn temperature is chosen from $\tau \in \{0.01, 0.05\}$, the input matrix to the neural MSS algorithm $\in \mathbb{R}^{N \times N}$, and $\boldsymbol{S}\boldsymbol{W}\boldsymbol{S}^\top[i, j] \in \mathbb{R}^{d'}$ where $d' = 1$. For the neural relaxation of the MSS algorithm, $\text{DPmsg}_\psi$ and $\text{DPupdate}_\psi$ are 2-layer feedforward networks with ReLU activation mapping from $2d'$ to $d'$. Meanwhile, $\text{DPreadout}_\psi$ is a 2-layer feedforward network with ReLU activation mapping from $d'$ to 1, which outputs the predicted clique number. For the design of $\mathcal{L}_{\text{Composite}}$, we set $\delta = 1$, and search for $\gamma, \lambda$ in $\{0.25, 0.75\}$, and $\{0.1, 1\}$ respectively. Hence, the total search space for hyperparameters consists of 8 combinations ($\{\tau\} \times \{\gamma\} \times \{\lambda\}$), and we select the best model out of the 8 hyperparameter configurations.

## D.2  DETAILS ABOUT OUR TRAINING

For the early stopping criteria based on the validation MSE, we use a patience parameter as 200 epochs. All models are trained using the Adam optimizer, with a learning rate of $10^{-3}$, and weight decay $5 \times 10^{-4}$.

## D.3  COMPLEXITY ANALYSIS OF MXNET

Considering $|V(G)| = N$, we compute the computational complexity of MXNET (MSS) and MXNET (SubMatch) as follows:

- MXNET (MSS): Since $\boldsymbol{X} \in \mathbb{R}^{N \times d}$, the complexity of computing $\boldsymbol{W} = \boldsymbol{A} \odot \boldsymbol{X}\boldsymbol{X}^\top$ is $O(N^2 d)$. Further, the computation of $\boldsymbol{S}$ has complexity $O(N^2)$. Since $\boldsymbol{S}$ approximates a permutation matrix, and the temperature $\lambda$ is small, we have a constant order of elements in each row and column leading to an $O(N^2)$ complexity for computing $\boldsymbol{S}\boldsymbol{W}\boldsymbol{S}^\top$. Finally, Algorithm 1 runs for $N^2$ iteration, giving an overall complexity of $O(N^2 d)$.
- MXNET (SubMatch): Since $\boldsymbol{X} \in \mathbb{R}^{N \times d}$, and $\boldsymbol{S}$ is sparse, computation of $\boldsymbol{S}\boldsymbol{X}$ has complexity $O(Nd)$ and we can compute $\sum_{i,j}[\boldsymbol{Q}_c - \boldsymbol{S}\boldsymbol{X}]_+[i, j]$ in $O(N)$ time. However, due to extensive tensorization in our implementations, the effective time complexity is much less. In Table 6, we compare the time (in ms) for a forward pass for different values of $N = |V|$.

Table 6: Time (in ms) for a forward pass for MXNET (SubMatch) on graphs with increasing $|V|$ from 100 to 900.

| $|V|$ | 100 | 200 | 300 | 400 | 500 | 600 | 700 | 800 | 900 |
|---|---|---|---|---|---|---|---|---|---|
| Time (ms) | 21.78 | 20.94 | 21.58 | 21.47 | 21.94 | 22.12 | 21.83 | 22.29 | 23.11 |

Further, note that since we store the adjacency matrices and the FF matrix from Algorithm 1 explicitly, our space complexity is $O(N^2)$.

## D.4 DETAILS ABOUT BASELINES

For all methods, we set the number of GNN propagation layers to $L = 5$, and the embedding dimension to $d = 10$. We use the default implementations publicly available on GitHub for all methods – (1) EGN [*], (2) SCT [*], (3) Difusco [*], (4) GFnet [*], (5) VAG-CO [*], and (6) SFE, NSFE, ST, Reinforce [*].

## D.5 HARDWARE AND LICENSE

We implement our models using Python 3.10 and PyTorch 2.3.0. The training of our models and the baselines was performed on servers containing AMD EPYC 7642 48-Core Processors at 2.30GHz CPUs, and Nvidia RTX A6000 GPUs. Running times of all methods are compared on a clean machine with the above specifications.

# E DETAILS ABOUT EXPERIMENTAL SETUP

## E.1 DATASETS DESCRIPTION

We perform comprehensive experiments on eight datasets, including five real-world and three synthetic datasets. For the real-world datasets, following (Sanokowski et al., 2023), we use (1) IMDB-BINARY (IMDB), (2) ENZYMES (Enzymes) and modular product graphs from three real-world datasets sourced from the TUDatasets repository (Morris et al., 2020): (3) PTC-MM, (4) AIDS and (5) Mutagenicity. We use PTC-MM-$m$, AIDS-$m$ and MUTAG-$m$ to denote these datasets consisting of *m*odular products. The clique number in these graphs represents the size of the maximum common induced subgraph. Given graph $G_1, G_2$, the modular product graph $H = G_1 \diamond G_2$ has $V(H) = V(G_1) \times V(G_2)$, and nodes $(u_1, u_2), (v_1, v_2) \in V(H)$ have an edge if (i) $u_1 = v_1 \wedge (u_2, v_2) \in E(G_2)$, (ii) $u_2 = v_2 \wedge (u_1, v_1) \in E_{(}G_1)$, (iii) $(u_1, v_1) \in E(G_1) \wedge (u_2, v_2) \in E(G_2)$, or (iv) $(u_1, v_1) \notin E(G_1) \wedge (u_2, v_2) \notin \wedge E(G_2)$. It is known that $\omega(H)$ is the size of the maximum common induced subgraph of $G_1$ and $G_2$. In addition to these five real datasets, we also generate three synthetic datasets— (6) DSJC (Johnson & Trick, 1996), consisting of $k$-partite random graphs with at least one $k$-clique; (7) Brockington (Johnson & Trick, 1996), consisting of graphs with cliques hidden within low-degree nodes; and (8) RB (RB) (Xu et al., 2007), representing random hard constraint satisfaction problem instances that yield hard-to-find cliques. We generate one dataset consisting of 50 nodes sized graphs from DSJC (DSJC) and Brockington (Brock) families. We use the Gurobi solver (Gurobi Optimization, LLC, 2023) (with no time-limit) to generate the gold ground truth clique numbers for distant supervision.

## E.2 DATASETS STATISTICS

Table 7: Description of the datasets used for training and evaluation on clique number estimation. Table reports number of graphs, average and maximum number of nodes ($|V|$), average and maximum number of edges $|E|$, average edge density ($|E|/\binom{|V|}{2}$) and the average and maximum ground truth clique numbers ($\omega$).

| | Num Graphs | Avg. $|V|$ | Max. $|V|$ | Avg. $|E|$ | Max. $|E|$ | Avg. $|E|/\binom{|V|}{2}$ | Avg. $\omega$ | Max. $\omega$ |
|---|---|---|---|---|---|---|---|---|
| IMDB (Morris et al., 2020) | 537 | 23.382 | 136 | 106.317 | 1249 | 0.413 | 9.724 | 30 |
| Brock (Johnson & Trick, 1996) | 1000 | 50.000 | 50 | 857.029 | 1225 | 0.700 | 24.875 | 50 |
| DSJC (Johnson & Trick, 1996) | 1000 | 50.000 | 50 | 730.820 | 936 | 0.597 | 20.640 | 36 |
| AIDS-$m$ (Morris et al., 2020) | 1000 | 96.039 | 110 | 2470.012 | 3528 | 0.523 | 7.678 | 10 |
| PTC-MM-$m$ (Morris et al., 2020) | 2372 | 79.847 | 100 | 1518.133 | 2700 | 0.453 | 5.962 | 9 |
| Enzymes (Morris et al., 2020) | 595 | 32.482 | 125 | 62.168 | 149 | 0.160 | 3.797 | 5 |
| MUTAG-$m$ (Morris et al., 2020) | 1000 | 258.435 | 300 | 22352.642 | 31654 | 0.660 | 11.203 | 15 |
| RB (Xu et al., 2007) | 1000 | 252.276 | 300 | 4069.608 | 8805 | 0.126 | 17.428 | 26 |

## E.3 FURTHER NOTES ON HEAVY-LIFTING DECODERS

Our experiments suggest that some decoders in prior methods are very powerful, capable of returning near-maximal cliques starting from even uninformative heatmaps from naive heuristics. For some datasets, just sorting the nodes in decreasing order of degree or PageRank, and then including nodes

---

[*] https://github.com/Stalence/erdos_neu
[*] https://github.com/yimengmin/GeometricScatteringMaximalClique
[*] https://github.com/Edward-Sun/DIFUSCO
[*] https://github.com/zdhNarsil/GFlowNet-CombOpt
[*] https://github.com/ml-jku/VAG-CO
[*] https://github.com/Stalence/NeuralExt

that form a clique with already-included nodes, performs quite well. In some cases, removing the decoder and estimating the clique number directly from the heatmap resulted in drastic degradation of accuracy. This is a testament to how much value the decoder adds to the overall method, compared to the module that computes the heatmap. Some decoders involve backtracking search over node selection decisions informed by the heatmap. Some of the decoders simply consist of repeated sampling from the heatmap distribution and checking if the sample forms a clique — with such decoders, performance increases monotonically with increased investment of decoding time. In some implementations, sampling more costs predictably more time. In others, clever tensorization masks such costs effectively. This makes time-accuracy comparisons of non-standardized implementations based on wall-clock time quite misleading.

### E.4    FURTHER NOTES ON EVALUATION METRICS

We have proposed (R)MSE as a robust measure of estimation quality. In the combinatorial optimization literature, it is common to measure the approximation ratio between the reported clique size and the ground truth clique size, which is at most 1 (for methods that find actual cliques), averaged over instances. Approximation ratio is not appropriate for methods that can both overestimate and underestimate the ground truth clique size, because estimation errors across multiple instances may cancel out, giving an excessively optimistic impression of a method.

### E.5    DISCUSSION ON BASELINES

In the context of maximum clique detection, methods can be broadly classified into supervised learning, unsupervised learning and reinforcement learning based approaches.

**Supervised under extreme supervision**: DIFUSCO (Sun & Yang, 2024) uses extreme supervision by incorporating the exact maximum clique as input features and employing Cross Entropy Loss for training. This method is paired with a non-differentiable greedy clique number decoder.

**Unsupervised**: Several methods estimate the clique number without supervision. Our experiments include: **(1)** SFE and NSFE (Karalias et al., 2022), which are end-to-end differentiable methods that extend set functions to continuous domains; **(2)** REINFORCE algorithm (Williams, 1992) and **(3)** Straight-Through (ST) Estimator (Bengio et al., 2013), which enable backpropagation through discrete functions; **(4)** EGN (Karalias & Loukas, 2020), which trains a GNN with a probabilistic penalty loss function and retrieves the clique number using a probabilistic-sampling-based decoder; and **(4)** Scattering GCN (SCT) (Min et al., 2022), which trains a scattering GNN with a probabilistic penalty loss and uses a non-differentiable walk-based decoder. **(5)** NeurSC: Neural Subgraph Counting with Wasserstein Estimator (NeurSC) NeurSC addresses the subgraph counting problem using neural networks with GNN-based estimators that leverage both intra-graph and inter-graph neural architectures, complemented by a Wasserstein discriminator. We adapted NeurSC as a baseline for our clique number prediction framework by employing increasing k-cliques as query graphs and determining the subgraph count for each. The clique number is then estimated as the largest value of $k$ for which a non-zero count is returned.

In its default implementation, NeurSC employs a neural network with a final ReLU layer to ensure non-negative count predictions. After experimenting with various loss functions, we found that the best performance is achieved by removing the final ReLU layer and training the model to output positive scores for query cliques with ( $k \leq \omega(G)$ ) the ground truth clique number and negative scores for ( $k > \omega(G)$ ) the ground truth. This adaptation simplifies inference by identifying the point in the sequence of query cliques where the predicted score transitions from positive to negative, while also enabling effective training.

**Reinforcement Learning**: We include: **(1)** VAG-CO (Sanokowski et al., 2023), which frames the clique number problem as a variational learning problem in an RL setting and uses a non-differentiable decoder to sample the clique number. **(2)** GFlowNet (GFNET) (Zhang et al., 2024), which designs an MDP with states forming a latent flow network and estimates the clique number through a non-differentiable heuristic-based decoder.

**Details about Decoder-reliant baselines** We observe that methods with a non-differentiable component in the neural pipeline rely on decoders to generate the clique number from the output of the model. We describe each of the baselines below:

1. Heuristic: This generates heatmap values for a graph– which may be based upon graph statistics such as node degree, clustering coefficient or pagerank . It first sorts these values in non-increasing

order and iterates over the nodes, adding them to the solution set if (i) the total number of edges in the clique increases, and (ii) the clique constraint is not violated. We use the greedy decoder with the Difusco (Cont) and Difusco (Cat) baselines, as described next.

2. Difusco: Difusco generates multimodal distribution heatmaps for the underlying task, with two variants: Difusco (Cont) for diffusion with continuous Gaussian noise, and Difusco (Cat) for diffusion with categorical Bernoulli noise. Originally evaluated on the traveling salesman problem and the maximum independent set (MIS), for which a greedy decoder is used, we implement a similar greedy decoder for the clique number problem and evaluate DIFUSCO on this task.

3. EGN: EGN outputs the probability of each node belonging to the maximum clique. The probabilistic decoder used by EGN sorts these probability values in non-increasing order and iteratively adds each node to the solution set if it decreases a constrained probabilistic loss function, which is zero when a clique is detected. The maximum over multiple solutions is reported as the clique number.

4. SCT: Similar to EGN, SCAT outputs a probability value for each node indicating its likelihood of being in the maximum clique. The SCAT decoder sorts these values in non-decreasing order and iterates over the nodes. At each step, it starts a walk on the remaining nodes, adding a node to the solution set if the clique property is satisfied. The maximum cardinality of all found solutions is returned.

5. VAG-CO: VAG-CO approaches to solve the clique number problem by framing it in the space of Ising Models, and trains it in an RL manner, and finally samples solutions from a learned autoregressive parameterized distribution

6. GFnet: Since GFnet is a RL method, it maintains states and actions. At each iteration step, it samples an action, which decides which node to include in the clique. If the node inclusion violates the clique condition in the solution, it is removed from the solution set. Moreover, after addition of a node, it does a degree-based pruning to remove any node which might not be feasible.

## F  ADDITIONAL EXPERIMENTS

### F.1  RESULTS ON COMPARISONS WITH STATE-OF-THE-ART BASELINES WITH STANDARD ERROR

In the main paper, Table 1 compares MXNET and MXNET (MSS), against all the state-of-the art baselines. We were unable to report standard error due to space constraints. In Table 8, we report the results along with the standard error on all eight datasets. The standard error is computed over the squared error over all of the graphs in each dataset.

Table 8: Comparison of MXNET against state-of-the-art baselines in terms of Mean Squared Error (MSE) with standard error (lower is better). The baselines include EGN (Karalias & Loukas, 2020), SCT (Min et al., 2022), Difusco (Cat) (Sun & Yang, 2024), Difusco (Cont) (Sun & Yang, 2024), GFnet (Zhang et al., 2024), VAG-CO (Sanokowski et al., 2023), SFE (Karalias et al., 2022), NSFE (Karalias et al., 2022), ST (Bengio et al., 2013), and Reinforce (Williams, 1992). Baselines are divided into two categories: (i) those with a decoder, and (ii) those without. Numbers in green, blue, and yellow indicate the best, second-best, and third-best performers, respectively.

| | | IMDB | AIDS-$m$ | PTC-MM-$m$ | DSJC |
|---|---|---|---|---|---|
| **Decoder** | EGN (Karalias & Loukas, 2020) | $0.102 \pm 0.054$ | $0.610 \pm 0.060$ | $0.284 \pm 0.026$ | $0.030 \pm 0.021$ |
| | SCT (Min et al., 2022) | $4.102 \pm 0.810$ | $3.865 \pm 0.278$ | $1.802 \pm 0.109$ | $92.675 \pm 8.394$ |
| | Difusco (Cont) (Sun & Yang, 2024) | $5.333 \pm 2.174$ | $4.665 \pm 0.277$ | $2.621 \pm 0.139$ | $59.820 \pm 6.776$ |
| | Difusco (Cat) (Sun & Yang, 2024) | $1.361 \pm 0.510$ | $4.170 \pm 0.289$ | $2.244 \pm 0.129$ | $1.560 \pm 0.473$ |
| | GFnet (Zhang et al., 2024) | $2.815 \pm 0.779$ | $0.740 \pm 0.085$ | $0.554 \pm 0.045$ | $4.820 \pm 1.388$ |
| | VAG-CO (Sanokowski et al., 2023) | $5.037 \pm 1.896$ | $33.905 \pm 0.800$ | $19.962 \pm 0.540$ | $7.665 \pm 0.529$ |
| **Non-Decoder** | SFE (Karalias et al., 2022) | $4.833 \pm 1.556$ | $44.340 \pm 0.853$ | $26.568 \pm 0.649$ | $12.465 \pm 1.803$ |
| | NSFE (Karalias et al., 2022) | $3.185 \pm 1.394$ | $36.005 \pm 0.911$ | $19.800 \pm 0.583$ | $5.255 \pm 1.048$ |
| | ST (Bengio et al., 2013) | $8.972 \pm 2.595$ | $44.885 \pm 0.882$ | $26.703 \pm 0.640$ | $5.410 \pm 0.991$ |
| | Reinforce (Williams, 1992) | $16.472 \pm 4.341$ | $59.170 \pm 1.014$ | $37.455 \pm 0.797$ | $13.280 \pm 3.884$ |
| | NeurSC (Wang et al., 2022) | $5.330 \pm 0.22$ | $1.240 \pm 0.079$ | $1.510 \pm 0.056$ | $15.350 \pm 0.268$ |
| | MXNET (`ES:MSS`) | $0.094 \pm 0.022$ | $0.341 \pm 0.044$ | $0.179 \pm 0.013$ | $0.383 \pm 0.074$ |
| | MXNET (`ES:Bi`) | $0.094 \pm 0.022$ | $0.341 \pm 0.044$ | $0.179 \pm 0.013$ | $0.383 \pm 0.074$ |
| | | Brock | Enzymes | RB | MUTAG-$m$ |
| **Decoder** | EGN (Karalias & Loukas, 2020) | $1.310 \pm 0.453$ | $0.109 \pm 0.041$ | $15.615 \pm 1.738$ | $1.010 \pm 0.101$ |
| | SCT (Min et al., 2022) | $35.885 \pm 5.868$ | $0.891 \pm 0.124$ | $50.230 \pm 3.489$ | $11.105 \pm 0.855$ |
| | Difusco (Cont) (Sun & Yang, 2024) | $39.040 \pm 12.402$ | $1.034 \pm 0.133$ | $63.490 \pm 4.258$ | $10.590 \pm 0.610$ |
| | Difusco (Cat) (Sun & Yang, 2024) | $10.595 \pm 2.584$ | $0.950 \pm 0.103$ | $55.630 \pm 4.118$ | $10.965 \pm 0.598$ |
| | GFnet (Zhang et al., 2024) | $8.975 \pm 1.574$ | $0.857 \pm 0.093$ | $34.200 \pm 2.989$ | $2.565 \pm 0.208$ |
| | VAG-CO (Sanokowski et al., 2023) | $15.450 \pm 2.186$ | $4.244 \pm 0.244$ | $320.240 \pm 7.867$ | $132.700 \pm 3.046$ |
| **Non-Decoder** | SFE (Karalias et al., 2022) | $53.595 \pm 6.124$ | $1.630 \pm 0.210$ | $72.020 \pm 3.002$ | $110.905 \pm 2.764$ |
| | NSFE (Karalias et al., 2022) | $47.920 \pm 5.543$ | $1.118 \pm 0.170$ | $78.890 \pm 4.220$ | $96.445 \pm 2.636$ |
| | ST (Bengio et al., 2013) | $55.740 \pm 6.363$ | $5.454 \pm 0.284$ | $131.230 \pm 6.584$ | $111.020 \pm 2.769$ |
| | Reinforce (Williams, 1992) | $403.095 \pm 37.037$ | $1.731 \pm 0.142$ | $229.435 \pm 10.409$ | $132.700 \pm 3.054$ |
| | NeurSC (Wang et al., 2022) | $88.790 \pm 0.667$ | $0.630 \pm 0.073$ | $10.080 \pm 0.214$ | $3.520 \pm 0.132$ |
| | MXNET (`ES:MSS`) | $10.022 \pm 1.185$ | $0.203 \pm 0.024$ | $6.875 \pm 0.604$ | $4.130 \pm 0.399$ |
| | MXNET (`ES:Bi`) | $10.022 \pm 1.185$ | $0.196 \pm 0.026$ | $6.875 \pm 0.604$ | $0.613 \pm 0.067$ |

### F.2 INVESTIGATING THE EFFECT OF BICRITERIA EARLY STOPPING

In Table 3, we compared the effect of training with the three different early stopping strategies, leading to MxNet (**ES:MSS**), MxNet (**ES:SubMatch**) and MxNet (**ES:Bi**). Due to space constraints, we could not report the standard error numbers in the main paper. Here, in Table 9 we present results in terms of MSE and standard error for all eight datasets.

Table 9: Performance of MxNet (Composite) for different early stopping criteria, which are based on $\mathcal{L}_{\text{MSS}}$, $\mathcal{L}_{\text{SubMatch}}$ and $\mathcal{L}_{\text{Composite}}$ respectively. For each of these early stopping criteria, we use MxNet (MSS) and MxNet (SubMatch) (denoted as MSS and SubMatch respectively) components to measure the performance in terms of MSE on the test set. Numbers in green indicate the best performers.

| | IMDB | | AIDS-$m$ | | PTC-MM-$m$ | | DSJC | |
|---|---|---|---|---|---|---|---|---|
| | MSS | SubMatch | MSS | SubMatch | MSS | SubMatch | MSS | SubMatch |
| MxNet(**ES:MSS**) | $0.094 \pm 0.022$ | $0.241 \pm 0.069$ | $0.341 \pm 0.044$ | $0.705 \pm 0.186$ | $0.179 \pm 0.013$ | $0.265 \pm 0.027$ | $0.383 \pm 0.074$ | $9.360 \pm 2.438$ |
| MxNet(**ES:SubMatch**) | $0.331 \pm 0.125$ | $0.065 \pm 0.040$ | $0.307 \pm 0.035$ | $0.450 \pm 0.052$ | $0.232 \pm 0.015$ | $0.495 \pm 0.039$ | $0.581 \pm 0.094$ | $7.335 \pm 1.407$ |
| MxNet(**ES:Bi**) | $0.094 \pm 0.022$ | $0.241 \pm 0.069$ | $0.341 \pm 0.044$ | $0.705 \pm 0.186$ | $0.179 \pm 0.013$ | $0.265 \pm 0.027$ | $0.383 \pm 0.074$ | $9.360 \pm 2.438$ |

| | Brock | | Enzymes | | RB | | MUTAG-$m$ | |
|---|---|---|---|---|---|---|---|---|
| | MSS | SubMatch | MSS | SubMatch | MSS | MSS | MSS | SubMatch |
| MxNet(**ES:MSS**) | $10.022 \pm 1.185$ | $726.995 \pm 43.564$ | $0.203 \pm 0.024$ | $0.353 \pm 0.073$ | $6.875 \pm 0.604$ | $35.620 \pm 9.273$ | $4.130 \pm 0.399$ | $7.805 \pm 0.989$ |
| MxNet(**ES:SubMatch**) | $19.165 \pm 1.839$ | $170.280 \pm 14.236$ | $0.196 \pm 0.026$ | $0.210 \pm 0.038$ | $10.167 \pm 0.879$ | $6.830 \pm 0.752$ | $4.151 \pm 0.392$ | $4.040 \pm 0.333$ |
| MxNet(**ES:Bi**) | $10.022 \pm 1.185$ | $726.995 \pm 43.564$ | $0.196 \pm 0.026$ | $0.210 \pm 0.038$ | $6.875 \pm 0.604$ | $35.620 \pm 9.273$ | $0.613 \pm 0.067$ | $4.680 \pm 0.602$ |

### F.3 ABLATION STUDY ON VARIANTS OF MxNet

In the main paper, Table 2 presents an ablation study where we compare the impact of the three possible variations of our models, *viz.* MxNet (MSS), MxNet (SubMatch), and MxNet (Composite) in terms of MSE. Due to space constraints, we did not report the standard error numbers in the main. We now do so in Table 10.

Table 10: Comparison of MxNet (MSS), MxNet (SubMatch), and MxNet variants in terms of Mean Squared Error (MSE) on eight datasets. Numbers in green, blue indicate the best performers and second best performers respectively.

| | IMDB | AIDS-$m$ | PTC-MM-$m$ | DSJC |
|---|---|---|---|---|
| MxNet (MSS) | $24.434 \pm 7.392$ | $0.295 \pm 0.035$ | $2.364 \pm 0.174$ | $66.890 \pm 5.394$ |
| MxNet (SubMatch) | $0.056 \pm 0.039$ | $0.420 \pm 0.049$ | $0.512 \pm 0.039$ | $2.820 \pm 0.772$ |
| MxNet | $0.094 \pm 0.022$ | $0.341 \pm 0.044$ | $0.179 \pm 0.013$ | $0.383 \pm 0.074$ |

| | Brock | Enzymes | RB | MUTAG-$m$ |
|---|---|---|---|---|
| MxNet (MSS) | $150.430 \pm 10.911$ | $0.205 \pm 0.023$ | $10.190 \pm 0.879$ | $4.130 \pm 0.399$ |
| MxNet (SubMatch) | $150.890 \pm 11.504$ | $0.210 \pm 0.038$ | $7.170 \pm 0.975$ | $4.040 \pm 0.333$ |
| MxNet | $10.022 \pm 1.185$ | $0.196 \pm 0.026$ | $6.875 \pm 0.604$ | $0.613 \pm 0.067$ |

F.4  ADAPTION OF BASELINES TO LEARNING UNDER DISTANT SUPERVISION

Here, we expand our exploration of supervised methods by adapting decoder-based baselines to our direct clique prediction objective. Although there are no clear guidelines for customizing decoder techniques specifically for our purposes, we outline our adaptation attempts for each baseline below.

**Extension to Non-decoder Unsupervised baselines**  We have four baselines based on the Set Function Extension (SFE) paradigm proposed by Karalias et al. (2022). These methods, while unsupervised and devoid of iterative decoding mechanisms, leverage SFEs to design loss functions centered on discrete objectives arising from the maximum clique detection task, which involves binary prediction of clique-affiliated nodes. To align this with our clique detection objective, we introduce a supervised mean square error (MSE) loss component to their original loss function, treating the predicted clique number as the summation of their predicted set indicator values. This combined loss function aims to minimize both the original SFE-based objective and the discrepancy between the predicted and ground truth clique-affiliated nodes. Consequently, we derive four supervised variants of the corresponding baselines: SFE-SUP, NSFE-SUP, Reinforce-SUP, and ST-SUP.

**EGN**  While the original implementation of EGN is unsupervised, we enhance it by providing a supervision signal derived from the ground truth clique number. First, we obtain the node-level heatmap output by EGN for a graph $G$, which indicates the probability $p_i$ of each node participating in the maximum clique. Then, we add an additional MSE loss term to their objective: $(\sum_i p_i - \omega(G)))^2$. During inference, we use the node-level heatmaps to threshold to binary (clique-participation) indicators and then sum them to obtain the predicted clique number as $\sum_i \mathbf{1}[p_i \geq 0.5]$, resulting in the supervised baseline EGN-SUP.

**SCT**  It uses a distinct scattering GCN-based encoder that also outputs a node-level heatmap. This heatmap is subsequently fed into their walk-based decoder. Following a similar approach as with EGN, we were able to incorporate supervision by adding an MSE loss term based on the predicted node participation probabilities in the maximum clique. This adaptation results in the supervised baseline SCT-SUP.

**Difusco**  They employ extreme supervision based on binary indicators of ground truth node-level clique participation. Their model generates multi-modal distribution heatmaps over nodes using either categorical (Difusco (Cat)) or Gaussian (Difusco (Cont)) noise, trained under cross-entropy loss. Subsequently, they use a greedy decoder. For our purposes, we allow them to continue using their extreme supervision pipeline but replace their decoder with a thresholding mechanism to obtain the clique number as $\sum_i \mathbf{1}[p_i \geq 0.5]$, where $p_i$ are the entries of the diffusion denoising output mechanism, normalized to values between 0 and 1. This approach gives rise to the two baselines Difusco (Cat)-SUP and Difusco (Cont)-SUP.

Table 11: Comparison of MxNET against all the baselines in terms of MSE. Here, the baselines are modified to be trained under distance supervision. We name each baseline as "baseline-SUP". Numbers in green, blue, and yellow indicate the best, second-best, and third-best performers, respectively.

| | IMDB | AIDS-$m$ | PTC-MM-$m$ | DSJC | Brock | Enzymes | RB | MUTAG-$m$ |
|---|---|---|---|---|---|---|---|---|
| EGN-SUP | 13.5556 | 18.9350 | 10.0589 | 1.7950 | 140.4650 | 43.9916 | 59.5200 | 493.0050 |
| SCT-SUP | 265.2037 | 5638.2300 | 2536.1389 | 440.0850 | 338.0600 | 101.5714 | 25617.3203 | 27464.9141 |
| Difusco (Cont)-SUP | 110.3426 | 1372.7999 | 1091.7263 | 432.9300 | 313.1450 | 304.0672 | 22398.1895 | 4736.3198 |
| Difusco (Cat)-SUP | 16.2870 | 59.1700 | 20.8716 | 10.9500 | 60.5550 | 9.9496 | 142.5900 | 132.7000 |
| SFE-SUP | 4.2222 | 44.7800 | 26.5116 | 12.3950 | 55.0400 | 2.0924 | 76.9500 | 110.8500 |
| NSFE-SUP | 4.3981 | 34.5100 | 21.0653 | 9.6850 | 47.2650 | 1.5966 | 88.5000 | 89.0150 |
| ST-SUP | 5.6111 | 44.4900 | 26.6337 | 5.1850 | 56.5700 | 5.5126 | 147.4450 | 110.8950 |
| Reinforce-SUP | 12.1667 | 59.1700 | 37.5221 | 11.7800 | 366.5650 | 1.6639 | 202.2800 | 132.7000 |
| MxNET | 0.094 | 0.341 | 0.179 | 0.383 | 10.022 | 0.196 | 6.875 | 0.613 |

Table 11 compares all these baselines in terms of MSE across eight datasets. We observe that:

**(1)** MxNET still outperforms all the supervised baselines by a significant margin. This is not surprising given that the baselines were not originally designed for the clique prediction task. Instead, they are optimized to provide a heatmap-based node ordering, which reduces the amortized solution generation time for their greedy decoder. However, we had no better alternative than attempting the approach as described.

**(2)** EGN-SUP maintains its position as "first among equals" by being the second best performer in four out of eight datasets, while NSFE-SUP is second best in three out of eight, and SFE-SUP is second best in the remaining one.

**(3)** In general, Difusco (Cat)-SUP and Difusco (Cont)-SUP are unable to match the performance of the other baselines, highlighting their significant dependence on the non-differentiable greedy decoding component.

**(4)** Reinforce-SUP is the third best performer in the Enzymes dataset and shows reasonable performance in the remaining datasets, although being behind EGN-SUP and NSFE-SUP.

### F.5 ZOOMING IN ON THE RESULTS OF COMPETITORS CLOSEST TO MXNET

Among all baselines, SFE, NSFE, ST Reinforce and NeurSC do not use a decoder and they are end-to-end differentiable. Hence, they are closest to our setup. Here we focus on them in isolation.

Table 12: Effect of distant supervision on unsupervised non-decoder baselines. Numbers in green indicate best performer overall, while numbers in blue indicate the second-best performer overall, and the better performer among the two variants per baseline.

|  | IMDB | AIDS-$m$ | PTC-MM-$m$ | DSJC | Brock | Enzymes | RB | MUTAG-$m$ |
|---|---|---|---|---|---|---|---|---|
| SFE | 4.833 | 44.340 | 26.568 | 12.465 | 53.595 | 1.630 | 72.020 | 110.905 |
| SFE (Supervised) | 4.222 | 44.780 | 26.512 | 12.395 | 55.040 | 2.092 | 76.950 | 110.850 |
| NSFE | 3.185 | 36.005 | 19.800 | 5.255 | 47.920 | 1.118 | 78.890 | 96.445 |
| NSFE (Supervised) | 4.398 | 34.510 | 21.065 | 9.6850 | 47.265 | 1.597 | 88.500 | 89.015 |
| ST | 8.972 | 44.885 | 26.703 | 5.410 | 55.740 | 5.454 | 131.230 | 111.020 |
| ST (Supervised) | 5.611 | 44.490 | 26.634 | 5.185 | 56.570 | 5.513 | 147.445 | 110.895 |
| Reinforce | 16.472 | 59.170 | 37.455 | 13.280 | 403.095 | 1.731 | 229.435 | 132.700 |
| Reinforce (Supervised) | 12.167 | 59.170 | 37.522 | 11.780 | 366.565 | 1.664 | 202.280 | 132.700 |
| MXNET | 0.094 | 0.341 | 0.179 | 0.383 | 10.022 | 0.196 | 6.875 | 0.613 |

We first look into the effect of supervision on end-to-end differentiable unsupervised baselines. Here, NeurSC, being a supervised method, is excluded from consideration. In Table 12, we compare the effect of additional supervision loss on the performance of these baselines: SFE vs SFE-SUP, NSFE vs NSFE-SUP, Reinforce vs Reinforce-SUP, and ST vs ST-SUP. We observe that:

**(1)** For SFE, Reinforce, and ST, the supervision leads to a moderate improvement in final clique prediction accuracy in approximately 60% of the cases.

**(2)** In NSFE, however, the supervision actually causes performance to drop in most cases. This suggests that the set function extension pipeline as currently leveraged is not very conducive to effective backpropagation-guided training through supervised loss terms.

In the main paper, Table 1 compares MXNET and MXNET (MSS) against all the state-of-the-art baselines, along with the four unsupervised end-to-end differentiable baselines. Due to space constraints, we were unable to report standard error or timing analysis. In Table 13, we provide the results along with the standard error, with the time in brackets alongside, across all eight datasets. The standard error is computed over the squared error of all graphs in each dataset. Brackets indicate the amortized running time per graph in seconds. We observe that:

**(1)** MXNET maintains a significant lead against all four baselines even when standard error is considered. Additionally, MXNET exhibits significantly lower standard error compared to all of the baselines.

**(2)** In terms of amortized run time, all four methods are comparable to, and often 2-3x faster than MXNET. This is due to their avoidance of any greedy decoding heuristic, resulting in extremely low inference time latency. However, despite MXNET being slightly behind them in terms of run-time, it still significantly outperforms them.

### F.6 ABLATIONS ON BASELINES AND DECODERS

We thoroughly investigate each of the decoder-reliant baselines to identify the contributions of their neural and decoder-based components to the clique detection and prediction task. The baselines, namely EGN, SCT, Difusco, and GFnet, each employ their own decoder mechanisms, warranting individual examination. Our ablation studies follow this theme:

**Increasing the number of parallel samples allowed** In contrast to Table 1 where each decoder-based baseline could propose only one clique and Table 4 where each decoder-based baseline could propose only one or four cliques, here we empower them to propose up to 8 cliques. We then take the maximum of these eight proposals as the predicted clique.

Table 13: Comparison of MxNET against state-of-the-art baselines, which do not use a decoder and are end-to-end differentiable, in terms of Mean Squared Error (MSE) with standard error (lower is better). Numbers highlighted in green, blue, and yellow indicate the best, second-best, and third-best performers, respectively. Brackets denote the amortized running time per graph in seconds. Best runtime (lowest inference latency) **bolded** for each dataset.

| | IMDB | AIDS-$m$ | PTC-MM-$m$ | DSJC |
|---|---|---|---|---|
| SFE | $4.833 \pm 1.556$ (0.013) | $44.340 \pm 0.853$ (0.013) | $26.568 \pm 0.649$ (0.009) | $12.465 \pm 1.803$ (0.010) |
| NSFE | $3.185 \pm 1.394$ (0.025) | $36.005 \pm 0.911$ (0.046) | $19.800 \pm 0.583$ (0.037) | $5.255 \pm 1.048$ (0.034) |
| ST | $8.972 \pm 2.595$ (0.013) | $44.885 \pm 0.882$ (0.014) | $26.703 \pm 0.640$ (0.009) | $5.410 \pm 0.991$ (0.009) |
| Reinforce | $16.472 \pm 4.341$ (**0.012**) | $59.170 \pm 1.014$ (**0.007**) | $37.455 \pm 0.797$ (**0.005**) | $13.280 \pm 3.884$ (**0.007**) |
| MxNET | $0.094 \pm 0.022$ (0.029) | $0.341 \pm 0.044$ (0.021) | $0.179 \pm 0.013$ (0.018) | $0.383 \pm 0.074$ (0.014) |
| | Brock | Enzymes | RB | MUTAG-$m$ |
| SFE | $53.595 \pm 6.124$ (0.009) | $1.630 \pm 0.210$ (0.013) | $72.020 \pm 3.002$ (0.024) | $110.905 \pm 2.764$ (0.024) |
| NSFE | $47.920 \pm 5.543$ (0.026) | $1.118 \pm 0.170$ (0.026) | $78.890 \pm 4.220$ (0.066) | $96.445 \pm 2.636$ (0.066) |
| ST | $55.740 \pm 6.363$ (0.009) | $5.454 \pm 0.284$ (0.013) | $131.230 \pm 6.584$ (0.020) | $111.020 \pm 2.769$ (0.020) |
| Reinforce | $403.095 \pm 37.037$ (**0.008**) | $1.731 \pm 0.142$ (**0.011**) | $229.435 \pm 10.409$ (**0.008**) | $132.700 \pm 3.054$ (**0.010**) |
| MxNET | $10.022 \pm 1.185$ (0.014) | $0.196 \pm 0.026$ (0.023) | $6.875 \pm 0.604$ (0.119) | $0.613 \pm 0.067$ (0.134) |

**Non-neural heuristic version** To isolate the impact of the decoder-based components, we design non-neural versions of these baselines. Here, we generate three alternative non-neural heatmaps based on elementary graph statistics: node degree, clustering coefficient, and pagerank. We provide each of these three resultant heatmaps in turn to each of the baseline decoder variants and evaluate their performance. We prefix the names of these non-neural baselines with Heuristic-●.

In the main paper, Table 1 compares MxNET and MxNET (MSS), against all the state-of-the art baselines. We were unable to report standard error due to space constraints. In Table 8, we report the results along with the standard error on all eight datasets. The standard error is computed over the squared error over all of the graphs in each dataset.

### F.6.1 ABLATION ON EGN

Table 14: Performance variation of EGN, across its non neural variants (Heuristic - ●) and different number of clique proposal samples used to select the clique. Results in terms of Mean Squared Error (MSE) with standard error (lower is better), along with amortized inference time per graph (in seconds). Numbers highlighted in green, blue, and yellow indicate the best, second-best, and third-best performers, respectively. Brackets denote the amortized running time per graph in seconds. Best runtime (lowest inference latency) **bolded** for each dataset.

| | IMDB | AIDS-$m$ | PTC-MM-$m$ | DSJC |
|---|---|---|---|---|
| EGN | $0.102 \pm 0.054$ (0.117) | $0.610 \pm 0.060$ (0.190) | $0.284 \pm 0.026$ (0.149) | $0.030 \pm 0.021$ (0.152) |
| EGN (4x) | $0.102 \pm 0.054$ (0.236) | $0.340 \pm 0.038$ (0.533) | $0.183 \pm 0.019$ (0.410) | $0.005 \pm 0.005$ (0.525) |
| EGN (8x) | $0.102 \pm 0.054$ (0.351) | $0.280 \pm 0.036$ (1.007) | $0.141 \pm 0.018$ (0.781) | $0.000 \pm 0.000$ (0.718) |
| Heuristic-Degree-EGN | $2.824 \pm 1.389$ (0.146) | $3.920 \pm 0.223$ (0.221) | $2.411 \pm 0.121$ (0.229) | $0.820 \pm 0.278$ (0.149) |
| Heuristic-Degree-EGN (4x) | $1.806 \pm 1.159$ (0.227) | $3.100 \pm 0.201$ (0.668) | $1.827 \pm 0.101$ (0.518) | $0.400 \pm 0.179$ (0.424) |
| Heuristic-Degree-EGN (8x) | $0.769 \pm 0.750$ (0.382) | $2.500 \pm 0.169$ (1.338) | $1.554 \pm 0.088$ (0.931) | $0.215 \pm 0.134$ (0.944) |
| Heuristic-ClusterCoeff-EGN | $3.750 \pm 0.787$ (0.079) | $1.500 \pm 0.139$ (0.198) | $0.861 \pm 0.074$ (0.159) | $5.490 \pm 1.680$ (0.138) |
| Heuristic-ClusterCoeff-EGN (4x) | $0.222 \pm 0.069$ (0.137) | $0.965 \pm 0.101$ (0.662) | $0.497 \pm 0.048$ (0.493) | $0.395 \pm 0.322$ (0.355) |
| Heuristic-ClusterCoeff-EGN (8x) | $0.065 \pm 0.040$ (0.271) | $0.675 \pm 0.079$ (0.939) | $0.366 \pm 0.038$ (0.698) | $0.035 \pm 0.022$ (0.668) |
| Heuristic-PageRank-EGN | $4.806 \pm 1.094$ (0.116) | $3.515 \pm 0.212$ (0.295) | $2.596 \pm 0.140$ (0.184) | $0.960 \pm 0.304$ (0.167) |
| Heuristic-PageRank-EGN (4x) | $2.139 \pm 0.898$ (0.257) | $2.755 \pm 0.190$ (0.672) | $2.080 \pm 0.120$ (0.491) | $0.490 \pm 0.204$ (0.418) |
| Heuristic-PageRank-EGN (8x) | $0.778 \pm 0.750$ (0.376) | $2.100 \pm 0.155$ (1.224) | $1.676 \pm 0.096$ (0.942) | $0.265 \pm 0.141$ (0.735) |
| MxNET | $0.094 \pm 0.022$ (**0.029**) | $0.341 \pm 0.044$ (**0.021**) | $0.179 \pm 0.013$ (**0.018**) | $0.383 \pm 0.074$ (**0.014**) |
| | Brock | Enzymes | RB | MUTAG-$m$ |
| EGN | $1.310 \pm 0.453$ (0.151) | $0.109 \pm 0.041$ (0.087) | $15.615 \pm 1.738$ (0.201) | $1.010 \pm 0.101$ (0.473) |
| EGN (4x) | $0.315 \pm 0.079$ (0.462) | $0.042 \pm 0.018$ (0.150) | $12.635 \pm 1.620$ (0.529) | $0.665 \pm 0.085$ (1.480) |
| EGN (8x) | $0.175 \pm 0.050$ (0.977) | $0.034 \pm 0.017$ (0.173) | $9.645 \pm 1.380$ (1.003) | $0.555 \pm 0.080$ (2.712) |
| Heuristic-Degree-EGN | $7.580 \pm 1.820$ (0.157) | $0.412 \pm 0.054$ (0.080) | $47.700 \pm 3.372$ (0.309) | $9.320 \pm 0.412$ (0.686) |
| Heuristic-Degree-EGN (4x) | $6.405 \pm 1.637$ (0.435) | $0.227 \pm 0.039$ (0.129) | $40.070 \pm 2.904$ (0.837) | $7.990 \pm 0.374$ (2.117) |
| Heuristic-Degree-EGN (8x) | $4.375 \pm 1.152$ (0.792) | $0.101 \pm 0.028$ (0.189) | $33.135 \pm 2.632$ (1.186) | $7.245 \pm 0.363$ (4.053) |
| Heuristic-ClusterCoeff-EGN | $14.900 \pm 6.377$ (0.151) | $0.143 \pm 0.032$ (0.069) | $61.270 \pm 2.933$ (0.198) | $4.475 \pm 0.289$ (1.110) |
| Heuristic-ClusterCoeff-EGN (4x) | $2.315 \pm 0.497$ (0.380) | $0.067 \pm 0.023$ (0.135) | $57.705 \pm 2.986$ (0.281) | $3.295 \pm 0.246$ (2.265) |
| Heuristic-ClusterCoeff-EGN (8x) | $1.360 \pm 0.394$ (0.715) | $0.034 \pm 0.017$ (0.196) | $54.740 \pm 3.004$ (0.425) | $2.875 \pm 0.220$ (3.647) |
| Heuristic-PageRank-EGN | $7.030 \pm 1.741$ (0.163) | $0.588 \pm 0.074$ (0.102) | $48.200 \pm 3.438$ (0.256) | $7.320 \pm 0.367$ (0.957) |
| Heuristic-PageRank-EGN (4x) | $6.215 \pm 1.646$ (0.422) | $0.202 \pm 0.037$ (0.127) | $39.665 \pm 2.853$ (0.673) | $6.670 \pm 0.348$ (2.307) |
| Heuristic-PageRank-EGN (8x) | $3.725 \pm 0.880$ (1.031) | $0.118 \pm 0.030$ (0.245) | $33.525 \pm 2.659$ (1.213) | $6.095 \pm 0.331$ (4.261) |
| MxNET | $10.022 \pm 1.185$ (**0.014**) | $0.196 \pm 0.026$ (**0.023**) | $6.875 \pm 0.604$ (**0.119**) | $0.613 \pm 0.067$ (**0.134**) |

In Table 14, we present the results of our ablation study in terms of Mean Squared Error (MSE) with standard error, along with per-graph amortized run-time in brackets. We evaluate three variants of the original baseline EGN with multiple samples: namely EGN, EGN (4x), and EGN (8x). Additionally, we examine three non-neural heuristic baselines based on node degree, clustering co-

efficient, and PageRank, each of which is allowed one, four, or eight samples. In the last row, we report the results for MXNET. We observe that:

**(1)** EGN shows significant improvement as the number of samples increases, with a 2-3x reduction in MSE in most cases. Notably, in Brock, there is an 8x improvement with the increase in samples.

**(2)** The performance improvement in EGN comes at the cost of increased inference time. For instance, in MUTAG-$m$, EGN (8x) results in almost 20X higher latency compared to MXNET, while providing only around a 50% improvement in MSE.

**(3)** The heuristic-driven baselines generally perform worse than their neural counterparts, highlighting the relative effectiveness of neural learning procedures.

**(4)** Although the neural baselines show slightly worse average MSE than the neural variants with decoders, they significantly outperform the decoder-free baselines. This underscores the significant enhancement in prediction accuracy achieved by using decoders, despite their non-differentiable nature.

### F.6.2 ABLATION ON SCT

Table 15: Performance variation of SCT, across its non neural variants (Heuristic - •) and different number of clique proposal samples used to select the clique. Results in terms of Mean Squared Error (MSE) with standard error (lower is better), along with amortized inference time per graph (in seconds). Numbers highlighted in green, blue, and yellow indicate the best, second-best, and third-best performers, respectively. Brackets denote the amortized running time per graph in seconds. Best runtime (lowest inference latency) **bolded** for each dataset.

| | IMDB | AIDS-$m$ | PTC-MM-$m$ | DSJC |
|---|---|---|---|---|
| SCT | 4.102 ± 0.810 (0.051) | 3.865 ± 0.278 (0.037) | 1.802 ± 0.109 (0.039) | 92.675 ± 8.394 (0.039) |
| SCT (4x) | 0.259 ± 0.077 (0.048) | 1.685 ± 0.146 (0.081) | 0.701 ± 0.058 (0.063) | 37.415 ± 4.525 (0.052) |
| SCT (8x) | 0.046 ± 0.020 (0.059) | 1.070 ± 0.108 (0.095) | 0.406 ± 0.037 (0.102) | 18.280 ± 2.676 (0.076) |
| Heuristic-Degree-SCT | 3.287 ± 1.410 (**0.002**) | 3.885 ± 0.220 (**0.008**) | 2.600 ± 0.125 (**0.005**) | 1.025 ± 0.312 (**0.003**) |
| Heuristic-Degree-SCT (4x) | 1.944 ± 1.342 (0.005) | 2.600 ± 0.172 (0.023) | 1.632 ± 0.088 (0.022) | 0.515 ± 0.186 (0.013) |
| Heuristic-Degree-SCT (8x) | 0.630 ± 0.177 (0.010) | 1.935 ± 0.144 (0.055) | 1.291 ± 0.075 (0.046) | 0.365 ± 0.150 (0.027) |
| Heuristic-ClusterCoeff-SCT | 10.556 ± 5.070 (**0.002**) | 1.545 ± 0.140 (0.030) | 0.922 ± 0.078 (0.016) | 5.490 ± 1.680 (0.008) |
| Heuristic-ClusterCoeff-SCT (4x) | 0.250 ± 0.100 (0.007) | 0.885 ± 0.093 (0.045) | 0.480 ± 0.049 (0.030) | 0.425 ± 0.322 (0.019) |
| Heuristic-ClusterCoeff-SCT (8x) | 0.019 ± 0.013 (0.012) | 0.555 ± 0.072 (0.076) | 0.364 ± 0.037 (0.046) | 0.185 ± 0.088 (0.025) |
| Heuristic-PageRank-SCT | 4.806 ± 1.094 (0.004) | 3.390 ± 0.208 (0.012) | 2.602 ± 0.141 (0.010) | 0.960 ± 0.304 (0.006) |
| Heuristic-PageRank-SCT (4x) | 1.028 ± 0.261 (0.008) | 2.310 ± 0.168 (0.026) | 1.785 ± 0.098 (0.028) | 0.470 ± 0.181 (0.015) |
| Heuristic-PageRank-SCT (8x) | 0.843 ± 0.197 (0.013) | 1.845 ± 0.145 (0.058) | 1.413 ± 0.075 (0.046) | 0.315 ± 0.144 (0.027) |
| MXNET | 0.094 ± 0.022 (0.029) | 0.341 ± 0.044 (0.021) | 0.179 ± 0.013 (0.018) | 0.383 ± 0.074 (0.014) |

| | Brock | Enzymes | RB | MUTAG-$m$ |
|---|---|---|---|---|
| SCT | 35.885 ± 5.868 (0.037) | 0.891 ± 0.124 (0.035) | 50.230 ± 3.489 (0.064) | 11.105 ± 0.855 (0.088) |
| SCT (4x) | 12.070 ± 2.144 (0.052) | 0.277 ± 0.051 (0.055) | 44.040 ± 2.875 (0.156) | 6.830 ± 0.558 (0.241) |
| SCT (8x) | 8.015 ± 1.452 (0.074) | 0.202 ± 0.047 (0.071) | 39.615 ± 2.716 (0.278) | 5.040 ± 0.450 (0.395) |
| Heuristic-Degree-SCT | 6.700 ± 1.648 (**0.004**) | 0.513 ± 0.068 (**0.002**) | 48.100 ± 3.309 (**0.017**) | 9.125 ± 0.392 (**0.040**) |
| Heuristic-Degree-SCT (4x) | 5.280 ± 1.438 (0.013) | 0.176 ± 0.035 (0.007) | 37.820 ± 2.899 (0.089) | 7.640 ± 0.357 (0.147) |
| Heuristic-Degree-SCT (8x) | 3.145 ± 0.802 (0.020) | 0.143 ± 0.032 (0.014) | 31.565 ± 2.767 (0.139) | 7.020 ± 0.334 (0.292) |
| Heuristic-ClusterCoeff-SCT | 20.340 ± 8.738 (0.009) | 0.134 ± 0.031 (0.003) | 62.950 ± 2.998 (0.055) | 4.930 ± 0.310 (0.544) |
| Heuristic-ClusterCoeff-SCT (4x) | 2.200 ± 0.496 (0.017) | 0.067 ± 0.023 (0.008) | 56.120 ± 3.138 (0.120) | 3.250 ± 0.244 (0.648) |
| Heuristic-ClusterCoeff-SCT (8x) | 1.755 ± 0.431 (0.031) | 0.017 ± 0.012 (0.018) | 52.550 ± 3.140 (0.201) | 2.750 ± 0.207 (0.793) |
| Heuristic-PageRank-SCT | 7.030 ± 1.741 (0.006) | 0.588 ± 0.074 (0.006) | 48.200 ± 3.438 (0.032) | 7.250 ± 0.370 (0.133) |
| Heuristic-PageRank-SCT (4x) | 4.545 ± 1.129 (0.016) | 0.151 ± 0.033 (0.010) | 37.485 ± 2.755 (0.095) | 6.535 ± 0.351 (0.239) |
| Heuristic-PageRank-SCT (8x) | 3.190 ± 0.770 (0.028) | 0.101 ± 0.028 (0.017) | 31.315 ± 2.588 (0.166) | 5.775 ± 0.317 (0.390) |
| MXNET | 10.022 ± 1.185 (0.014) | 0.196 ± 0.026 (0.023) | 6.875 ± 0.604 (0.119) | 0.613 ± 0.067 (0.134) |

In Table 15, we present the results of our ablation study in terms of Mean Squared Error (MSE) with standard error, along with per-graph amortized run-time in brackets. We evaluate three variants of the original baseline SCT with multiple samples: namely SCT, SCT (4x), and SCT (8x). Additionally, we examine three non-neural heuristic baselines based on node degree, clustering coefficient, and PageRank, each of which is allowed one, four, or eight samples. In the last row, we report the results for MXNET. We observe that:

**(1)** SCT demonstrates significant performance improvement with an increase in the number of samples. In most cases, performance improves approximately 2x when going from one to four samples, and the improvement is even more pronounced with eight samples. Notably, in IMDB, the improvement from one to eight samples is almost 10x.

**(2)** In general, the inference time latency for SCT is very low, being almost comparable to MXNET for one sample.

**(3)** Generally, the performance of SCT is significantly worse than that of MXNET, except in cases where eight samples are used. Even then, notably in larger graphs like RB and MUTAG-$m$, the performance of SCT with eight samples is still around 5-10x lower than MXNET.

**(4)** Remarkably, non-neural heuristics in the SCT decoder, particularly when allowed eight samples along with the clustering coefficient heuristic, perform exceptionally well. This highlights the substantial power of the SCT decoder, which is almost comparable to its neural variant across most datasets.

### F.6.3  ABLATION ON DIFUSCO

In Table 16, we present the results of our ablation study in terms of Mean Squared Error (MSE) with standard error, along with per-graph amortized run-time in brackets. We evaluate three variants of the original baselines Difusco (Cat) and Difusco (Cont) with multiple samples: namely Difusco (Cat), Difusco (Cont), Difusco (Cat) (4x), Difusco (Cont) (4x), Difusco (Cat) (8x) and Difusco (Cat) (8x). Additionally, we examine three non-neural heuristic baselines based on node degree, clustering coefficient, and PageRank. It's important to note that a key aspect of Difusco is that the randomness in the sampling procedure arises from the diffusion noise generation process. The greedy decoder used by Difusco is deterministic and does not allow for multiple samples. Therefore, unlike other baselines, every non-neural heuristic has only one set of observations based on a single sample. In the last row, we report the results for MXNET.

Table 16: Performance variation of Difusco (Cat) and Difusco (Cont), across its non neural variants (Heuristic - •) and different number of clique proposal samples used to select the clique. Results in terms of Mean Squared Error (MSE) with standard error (lower is better), along with amortized inference time per graph (in seconds). Numbers highlighted in green, blue, and yellow indicate the best, second-best, and third-best performers, respectively. Brackets denote the amortized running time per graph in seconds. Best runtime (lowest inference latency) **bolded** for each dataset.

| | IMDB | AIDS-$m$ | PTC-MM-$m$ | DSJC |
|---|---|---|---|---|
| Difusco (Cat) | 1.361 ± 0.510 (0.789) | 4.170 ± 0.289 (0.705) | 2.244 ± 0.129 (0.690) | 1.560 ± 0.473 (0.686) |
| Difusco (Cat) (4x) | 0.231 ± 0.153 (0.722) | 1.865 ± 0.145 (0.837) | 0.979 ± 0.066 (0.799) | 0.175 ± 0.094 (0.771) |
| Difusco (Cat) (8x) | 0.185 ± 0.149 (0.766) | 1.345 ± 0.117 (1.006) | 0.644 ± 0.047 (0.923) | 0.045 ± 0.023 (0.862) |
| Difusco (Cont) | 5.333 ± 2.174 (0.745) | 4.665 ± 0.277 (0.657) | 2.621 ± 0.139 (0.654) | 59.820 ± 6.776 (0.646) |
| Difusco (Cont) (4x) | 0.120 ± 0.055 (0.798) | 1.830 ± 0.131 (0.782) | 1.133 ± 0.072 (0.772) | 8.220 ± 1.573 (0.713) |
| Difusco (Cont) (8x) | 0.000 ± 0.000 (0.711) | 1.345 ± 0.107 (0.942) | 0.705 ± 0.049 (0.883) | 2.605 ± 0.546 (0.822) |
| Heuristic-Degree-Difusco (Cat) | 2.741 ± 1.388 (**0.011**) | 4.200 ± 0.247 (0.056) | 2.703 ± 0.132 (0.047) | 0.840 ± 0.258 (0.022) |
| Heuristic-ClusterCoeff-Difusco (Cat) | 12.370 ± 6.667 (0.012) | 1.495 ± 0.136 (0.077) | 0.931 ± 0.079 (0.045) | 5.490 ± 1.680 (0.029) |
| Heuristic-PageRank-Difusco (Cat) | 4.806 ± 1.094 (0.017) | 3.455 ± 0.211 (0.061) | 2.617 ± 0.141 (0.037) | 0.960 ± 0.304 (0.024) |
| MXNET | 0.094 ± 0.022 (0.029) | 0.341 ± 0.044 (**0.021**) | 0.179 ± 0.013 (**0.018**) | 0.383 ± 0.074 (**0.014**) |

| | Brock | Enzymes | RB | MUTAG-$m$ |
|---|---|---|---|---|
| Difusco (Cat) | 10.595 ± 2.584 (0.796) | 0.950 ± 0.103 (0.789) | 55.630 ± 4.118 (0.906) | 10.965 ± 0.598 (0.827) |
| Difusco (Cat) (4x) | 4.820 ± 1.400 (0.746) | 0.311 ± 0.059 (0.715) | 28.595 ± 2.690 (1.090) | 6.095 ± 0.321 (2.193) |
| Difusco (Cat) (8x) | 2.625 ± 0.726 (0.846) | 0.151 ± 0.033 (0.782) | 22.070 ± 2.475 (1.626) | 4.815 ± 0.269 (3.970) |
| Difusco (Cont) | 39.040 ± 12.402 (0.759) | 1.034 ± 0.133 (0.637) | 63.490 ± 4.258 (0.792) | 10.590 ± 0.610 (0.780) |
| Difusco (Cont) (4x) | 10.185 ± 3.823 (0.719) | 0.286 ± 0.051 (0.814) | 36.450 ± 3.063 (1.030) | 3.965 ± 0.301 (2.097) |
| Difusco (Cont) (8x) | 7.355 ± 4.001 (0.954) | 0.118 ± 0.030 (0.763) | 14.555 ± 1.787 (1.603) | 3.245 ± 0.275 (4.271) |
| Heuristic-Degree-Difusco (Cat) | 6.855 ± 1.723 (0.021) | 0.605 ± 0.089 (**0.016**) | 48.555 ± 3.418 (0.099) | 9.490 ± 0.440 (**0.110**) |
| Heuristic-ClusterCoeff-Difusco (Cat) | 20.520 ± 8.762 (0.036) | 0.143 ± 0.032 (0.017) | 62.960 ± 3.001 (0.152) | 4.805 ± 0.313 (0.615) |
| Heuristic-PageRank-Difusco (Cat) | 7.030 ± 1.741 (0.025) | 0.588 ± 0.074 (0.026) | 48.200 ± 3.438 (0.148) | 7.200 ± 0.363 (0.203) |
| MXNET | 10.022 ± 1.185 (**0.014**) | 0.196 ± 0.026 (0.023) | 6.875 ± 0.604 (0.119) | 0.613 ± 0.067 (0.134) |

We observe that:

**(1)** Difusco (Cat) generally performs better than Difusco (Cont) across the board. This demonstrates that Difusco (Cat)'s Bernoulli noise generation process is most suitable for the task of predicting discrete labels indicating node participation in a maximum clique, confirming claims made by the original authors (Sun & Yang, 2024).

**(2)** Like other decoder-based baselines, both Difusco (Cat) and Difusco (Cont) exhibit significant performance improvements when more samples are allowed. In IMDB, Difusco (Cat) achieves an MSE of zero without a substantial increase in inference time latency.

**(3)** The increase in inference time latency from one to four to eight samples is not as significant for Difusco (Cat) and Difusco (Cont) compared to other decoder-based baselines. This is attributed to the greedy deterministic nature of the decoder, which does not introduce large overhead relative to the entire pipeline.

**(4)** However, the average inference time latency of both Difusco (Cat) and Difusco (Cont) is significantly worse than both other baselines and MXNET. This is due to the costly noise generation and denoising steps undertaken by Difusco.

**(5)** Overall, MxNet outperforms both Difusco (Cat) and Difusco (Cont) in most datasets while maintaining significantly lower inference latency. Notably, in MUTAG-$m$, one of the larger datasets, MxNet's MSE performance is almost 4x better than Difusco's best, while being almost 20X faster at the same time.

**(6)** The non-neural heuristic baselines of Difusco can perform significantly better (e.g., in DSJC) or significantly worse (e.g., in IMDB) than Difusco's best. In general, the neural variants tend to outperform the non-neural ones, providing confidence in the effectiveness of the neural approach.

### F.6.4 Ablation on GFnet

In Table 17, we present the results of our ablation study in terms of Mean Squared Error (MSE) with standard error, along with per-graph amortized run-time in brackets. We evaluate three variants of the original baseline GFnet with multiple samples: namely GFnet, GFnet (4x), and GFnet (8x). Additionally, we examine three non-neural heuristic baselines based on node degree, clustering coefficient, and PageRank, each tested with one, four, or eight samples. GFnet is a reinforcement learning-based method where actions are sampled from a learned policy, with samples drawn from multinomial distributions over logits. For non-neural baselines, we replace the logits with graph statistic-based heatmaps. In the last row, we report the results for MxNet.

Table 17: Performance variation of GFnet, across its non neural variants (Heuristic - ●) and different number of clique proposal samples used to select the clique. Results in terms of Mean Squared Error (MSE) with standard error (lower is better), along with amortized inference time per graph (in seconds). Numbers highlighted in green, blue, and yellow indicate the best, second-best, and third-best performers, respectively. Brackets denote the amortized running time per graph in seconds. Best runtime (lowest inference latency) **bolded** for each dataset.

| | IMDB | AIDS-$m$ | PTC-MM-$m$ | DSJC |
|---|---|---|---|---|
| GFnet | $2.815 \pm 0.779$ (0.032) | $0.740 \pm 0.085$ (**0.017**) | $0.554 \pm 0.045$ (**0.011**) | $4.820 \pm 1.388$ (0.034) |
| GFnet (4x) | $0.167 \pm 0.066$ (0.081) | $0.210 \pm 0.034$ (0.054) | $0.166 \pm 0.017$ (0.035) | $0.175 \pm 0.080$ (0.109) |
| GFnet (8x) | $0.028 \pm 0.016$ (0.155) | $0.125 \pm 0.023$ (0.100) | $0.067 \pm 0.012$ (0.068) | $0.080 \pm 0.050$ (0.206) |
| Heuristic-Degree-GFnet | $2.333 \pm 1.351$ (0.018) | $3.865 \pm 0.223$ (0.035) | $2.623 \pm 0.130$ (0.027) | $1.170 \pm 0.351$ (0.032) |
| Heuristic-Degree-GFnet (4x) | $0.213 \pm 0.153$ (0.053) | $2.070 \pm 0.141$ (0.144) | $1.539 \pm 0.086$ (0.095) | $0.175 \pm 0.072$ (0.133) |
| Heuristic-Degree-GFnet (8x) | $0.806 \pm 0.750$ (0.098) | $1.660 \pm 0.115$ (0.281) | $1.105 \pm 0.069$ (0.189) | $0.175 \pm 0.080$ (0.243) |
| Heuristic-ClusterCoeff-GFnet | $8.759 \pm 4.564$ (**0.014**) | $4.690 \pm 0.274$ (0.059) | $2.817 \pm 0.163$ (0.034) | $134.200 \pm 10.512$ (0.034) |
| Heuristic-ClusterCoeff-GFnet (4x) | $0.361 \pm 0.133$ (0.056) | $1.865 \pm 0.140$ (0.234) | $1.160 \pm 0.074$ (0.143) | $29.815 \pm 3.700$ (0.111) |
| Heuristic-ClusterCoeff-GFnet (8x) | $0.083 \pm 0.042$ (0.087) | $1.280 \pm 0.101$ (0.456) | $0.731 \pm 0.050$ (0.283) | $11.290 \pm 1.921$ (0.241) |
| Heuristic-PageRank-GFnet | $16.213 \pm 6.751$ (0.018) | $4.710 \pm 0.285$ (0.040) | $2.646 \pm 0.136$ (0.028) | $144.875 \pm 11.211$ (0.028) |
| Heuristic-PageRank-GFnet (4x) | $0.343 \pm 0.133$ (0.064) | $1.970 \pm 0.141$ (0.159) | $1.248 \pm 0.080$ (0.107) | $38.695 \pm 4.577$ (0.103) |
| Heuristic-PageRank-GFnet (8x) | $0.083 \pm 0.042$ (0.107) | $1.300 \pm 0.105$ (0.312) | $0.718 \pm 0.053$ (0.202) | $11.165 \pm 1.908$ (0.193) |
| MxNet | $0.094 \pm 0.022$ (0.029) | $0.341 \pm 0.044$ (0.021) | $0.179 \pm 0.013$ (0.018) | $0.383 \pm 0.074$ (**0.014**) |
| | **Brock** | **Enzymes** | **RB** | **MUTAG-$m$** |
| GFnet | $8.975 \pm 1.574$ (0.040) | $0.857 \pm 0.093$ (0.016) | $34.200 \pm 2.989$ (**0.029**) | $2.565 \pm 0.208$ (**0.070**) |
| GFnet (4x) | $3.625 \pm 1.157$ (0.148) | $0.193 \pm 0.036$ (0.032) | $21.675 \pm 2.178$ (0.103) | $1.055 \pm 0.100$ (0.290) |
| GFnet (8x) | $1.125 \pm 0.328$ (0.289) | $0.050 \pm 0.020$ (0.059) | $17.745 \pm 2.060$ (0.203) | $0.610 \pm 0.072$ (0.578) |
| Heuristic-Degree-GFnet | $9.545 \pm 2.113$ (0.036) | $0.706 \pm 0.083$ (**0.009**) | $49.065 \pm 3.533$ (0.068) | $8.690 \pm 0.382$ (0.256) |
| Heuristic-Degree-GFnet (4x) | $3.110 \pm 0.822$ (0.141) | $0.227 \pm 0.039$ (0.027) | $39.565 \pm 2.963$ (0.273) | $6.580 \pm 0.325$ (1.012) |
| Heuristic-Degree-GFnet (8x) | $2.985 \pm 0.922$ (0.282) | $0.101 \pm 0.028$ (0.048) | $37.150 \pm 2.802$ (0.503) | $5.770 \pm 0.285$ (2.020) |
| Heuristic-ClusterCoeff-GFnet | $85.215 \pm 13.559$ (0.045) | $0.697 \pm 0.088$ (0.009) | $83.235 \pm 4.779$ (0.104) | $11.995 \pm 0.577$ (0.763) |
| Heuristic-ClusterCoeff-GFnet (4x) | $16.080 \pm 3.083$ (0.150) | $0.244 \pm 0.049$ (0.028) | $43.050 \pm 3.504$ (0.392) | $7.180 \pm 0.379$ (3.014) |
| Heuristic-ClusterCoeff-GFnet (8x) | $4.720 \pm 0.876$ (0.305) | $0.160 \pm 0.045$ (0.055) | $31.055 \pm 2.860$ (0.811) | $5.355 \pm 0.282$ (6.022) |
| Heuristic-PageRank-GFnet | $94.455 \pm 12.817$ (0.036) | $0.840 \pm 0.098$ (0.012) | $80.715 \pm 4.718$ (0.073) | $12.290 \pm 0.604$ (0.339) |
| Heuristic-PageRank-GFnet (4x) | $15.455 \pm 3.640$ (0.134) | $0.361 \pm 0.079$ (0.036) | $47.680 \pm 3.746$ (0.285) | $6.865 \pm 0.385$ (1.311) |
| Heuristic-PageRank-GFnet (8x) | $6.110 \pm 1.368$ (0.272) | $0.143 \pm 0.032$ (0.073) | $29.160 \pm 2.806$ (0.549) | $4.770 \pm 0.262$ (2.671) |
| MxNet | $10.022 \pm 1.185$ (**0.014**) | $0.196 \pm 0.026$ (0.023) | $6.875 \pm 0.604$ (0.119) | $0.613 \pm 0.067$ (0.134) |

Key observations include:

**(1)** Like earlier decoder-based methods, GFnet shows significant performance boosts with more samples. In most datasets, performance at least doubles, with the most notable improvement in DSJC, showing almost a 10x increase.

**(2)** The performance improvement coincides with increased inference latency. With one sample, GFnet is either comparable to or faster than MxNet. However, with eight samples, GFnet becomes significantly slower than MxNet in most cases.

**(3)** The non-neural variants of GFnet show similar MSE performance vs. inference time latency trade-offs. Among these, the node-degree heuristic performs best, sometimes ranking second overall.

**(4)** GFnet with eight samples often outperforms MxNet significantly. However, the inference time for GFnet is much higher. This highlights the trade-off between prediction accuracy and latency,

with decoder-based models iteratively deciding node inclusion in a clique for clique number prediction, compared to a direct prediction neural network like MXNET.

## F.7 ABLATION STUDY

(1) **Benefits of relaxing binary adjacency matrix to continuous values:** In the following, we show the results of ablation study.

Table 18: Effects of relaxing binary adjacency matrix to continuous values

|  | IMDB | PTC-MM-$m$ | DSJC |
|---|---|---|---|
| MXNET | 0.094 | 0.179 | 0.383 |
| Binary $A$ in MXNET | 86.231 | 2.364 | 0.470 |

We observe that smooth surrogate works better. We believe that there are two reasons behind this:

(i) CLIQUE is a classical hard-to-approximate problem in the worst case, so the promise of neural methods is to take advantage of distributions from which graphs are generated in an application. A natural way toward this goal is to train node representations $x_u$ that simultaneously predict the graph $A$ (via a generic GNN that aligns $x_u^\top x_v$ with $A[u, v]$), and make clique number prediction feasible (through a suitable task-customized network) — in a graph containing a $k$-clique that includes nodes $u$ and $v$, they share the same $k-1$ node neighborhood, leading to a high $x_u^\top x_v$ value.

(ii) In MXNET (MSS), the goal is to find node permutation $S$ that brings all 1s in $SAS^\top$ into a dense submatrix. Achieving this requires moving the zeros to the right places. When $S$ is relaxed to continuous values, non-zeros spread throughout $SAS^\top$, making it difficult to detect the MSS. Various thresholds or nonlinearities may reduce the problem, but also attenuate gradient signals when $x_u^\top x_v$ is very high or very low. Thus, $x_u^\top x_v$ scores serve as auxiliary signals for clique detection, but simply adding them to $A$ contributes much noise in the estimate of $\omega$. In contrast, the form $S(A \odot XX^\top)S^\top$ is better at absorbing the signal from $x_u$ without succumbing to its noise.

(iii) As a further mechanistic discussion, suppose, instead of $A \odot XX^\top$, we use $A$. Then, for each parameter $\theta \in \mathbb{R}$, the gradient term contains $SA\frac{dS^\top}{d\theta} + \frac{dS}{d\theta}AS^\top$. Here $S$ contains soft-max-alike-terms and therefore, $\frac{dS}{d\theta}$ will have terms such as $S[u, u']S[v, v']$. Thus, $SA\frac{dS^\top}{d\theta}$ consists of terms of the form $S[u, u']S[v, v']S[w, w']$, with each $S[\bullet, \bullet] \in [0, 1]$. This leads to attenuation of gradient signals. If we keep $A \odot XX^\top$, then the gradient will have $S[u, u']S[v, v']\left[\frac{dXX^\top}{d\theta}\right]_{u,u'}$, which will allow more significant gradient signals. Moreover, due to continuous adjacency entries, $S[u, u']S[v, v']S[w, w']$ will now be replaced with $S[u, u']S[v, v']S[w, w']X[u'', i]X[v'', j]$, where $X$s can control the gradient signals better, leading to actual learning of clique number computation for a distribution over graphs.

(2) **Relaxing Algorithm 1 using message passing GNN**: $MSS(B)$ can accurately compute clique only when $B$ has (close to) binary values. However our computation of $SWS^T$ involves general values in $(0, 1)$ from $S$ and +ve/-ve from $W$, which makes the entries of $B[i, j]$ to be +ve/-ve real numbers. Further, multiplication by $B[i, j]$ in line 7 in Algorithm 1 compounds this issue. To reduce these deleterious effects of fractional numbers on clique detection, several thresholding/clipping functions, such as Sigmoid, Tanh, ReLU, and ReLU6, may be applied, achieving varying degrees of success (we tried these). However, the best results were obtained by our final GNN-inspired solution, which could implement some form of local adaptive soft thresholding, as the message wavefront progressed. Note that this GNN is extremely lightweight, consisting of only 20 parameters.

Below, we compare the results of the best non-GNN thresholding variant (ReLU1 = $\min(\max(0, x), 1)$) against the results of MSS with the GNN-based Algorithm 1 implementation variant across three representative datasets of varying sizes: Enzymes , PTC-MM-$m$, and DSJC on average.

Table 19: Ablation of message passing GNN for Algorithm 1

|  | IMDB | PTC-MM-$m$ | DSJC |
|---|---|---|---|
| MXNET | 0.094 | 0.179 | 0.383 |
| No GNN in Algorithm 1 | 1.970 | 0.254 | 0.997 |

We observe that using a GNN based implementation for Algorithm 1 affords a performance improvement across all datasets.

(3) **Relaxing clique coverage loss in MxNet**: $\min_{\boldsymbol{P}}[\boldsymbol{K}_c - \boldsymbol{P}\boldsymbol{A}\boldsymbol{P}^\top]_+$ is a quadratic assignment problem (QAP), which is a hard problem to solve. Moreover, the binary values of $\boldsymbol{A}$ and $\boldsymbol{K}_c$ attenuate the gradient signals as described in the last para of item (1) above. We overcome the above challenges by replacing $\min_{\boldsymbol{P}}[\boldsymbol{K}_c - \boldsymbol{P}\boldsymbol{A}\boldsymbol{P}^\top]_+$ with $[\boldsymbol{Q}_c - \boldsymbol{S}\boldsymbol{X}]_+$. Instead of solving QAP, it solves a linear assignment problem, which is more tractable; and allows substantial gradient signals.

Table 20: Effect of relaxing coverage loss in MxNet (SubMatch)

|  | IMDB | PTC-MM-$m$ | DSJC |
|---|---|---|---|
| MxNet | 0.094 | 0.179 | 0.383 |
| Hard Clique Loss | 0.540 | 0.371 | 0.566 |

We present our results as follows, which shows smooth surrogate performs better.

## F.8 Performance on large graphs

There are two reasons why our primary focus was not on large graphs in the initial version. However, we have now stress-tested our method with additional experiments, as described later.

(1) Our goal is to design an end-to-end differentiable model— which is guided by the needs of practical applications like estimation of size of maximum induced common subgraph for graph simialrity. These applications typically involves fewer than 200 nodes (but possibly very many graphs).

Still, some of our datasets are much larger than the datasets used by the baselines. Specifically, our method uses graphs with upto 33K edges (Mutag-m), which is larger than all other datasets used in any end-to-end differentiable neural network based method. Moreover, we use the RB dataset, which is known to be notoriously hard for clique detection. Since clique number computation stands out as one of the hardest NP-complete problems (Hastad, 1996; Zuckerman, 2006), designing neural methods for even such moderate sized graphs is already quite challenging. It is unsurprising that neural methods have all limited themselves to relatively small graphs.

(2) Like any other ML task, neural clique detectors need *multiple* graphs for training and one entire graph represents one instance of data. Data sets with a single large graph are more common than data sets containing multiple large graphs with known clique numbers, which prevents a neural method to test its performance on large graphs. Nevertheless, we have the following proposal to predict clique number for a large graph, from which all neural models can benefit.

We decompose a large graph $\mathcal{G}$ into overlapping subgraphs $G_1, .., G_N$ each with moderate sizes. Then, the clique number of $\mathcal{G}$ is estimated as $\max_i \omega(G_i)$. This decomposition is helpful because (1) ground truth clique number generation is much easier in smaller graphs and (2) a neural clique detector requires multiple (graph, clique number) pairs for training. The risk of missing a clique straddling multiple subgraphs can be reduced via various degree based or K-core decomposition based heuristics.

Here, we worked with Amazon and email-Enron datasets. Amazon has 334,863 nodes and 925,872 edges. email-Enron has 36,692 nodes and 183,831 edges. The table below shows the true and predicted clique numbers.

Table 21: Results on large dataset

| Dataset | True clique number | Predicted clique number |
|---|---|---|
| Amazon | 7 | 6 |
| Email-enron | 20 | 19 |

We note that in each dataset, (1) MxNet was trained on a small set of subgraphs, sampled from the available single large graph, with corresponding ground truth clique numbers found using the Gurobi solver. During inference, MxNet effectively generalized to a much larger set of subgraphs extracted using degree-based heuristics, to ensure more comprehensive coverage of the large graph; (2) MxNet correctly identifies the subgraph containing the largest clique of the whole graph, i.e., $\arg\max_i \omega(G_i) = \arg\max_i \widehat{\omega}(G_i)$; (3) MxNet correctly predicts most of the member nodes of the maximum clique within the graph; (4) MxNet's predicted clique number for the large graph closely approximates the true clique number — much better than the predictions made by baselines, even with smaller graphs.

If the number of nodes $|V|$ is large, the entries of doubly stochastic matrix $S$ become more diffused. For the same temperature, the highest value in each row/column of $S$ becomes smaller as we increase $|V|$. Hence, we can decrease the temperature to ensure that $S$ has sufficiently low entropy to be interpretable. Further, we can use reparameterization of weight matrices (Zhai et al., 2023) in Sinkhorn network using spectral norm based scaling to facililate training in large graph training.

## F.9 ROBUSTNESS WITH RESPECT TO $\lambda$

Our experiments show that the proposed method is robust to lambda, as shown below.

Table 22: Variation of MSE with $\lambda$ for PTC-MM-$m$

| 0.221 | 0.283 | 0.172 | 0.179 | 0.163 | 0.402 | 1.456 |
|-------|-------|-------|-------|-------|-------|-------|

## F.10 FULL CURRICULUM TERMS VS. CURRICULUM LEARNING WITH FEWER TERMS

We performed experiments where in Eq. (9), we limited the first inner summation to only $c = \omega(G)$ and the second summation to only $c = \omega(G) - 1$. The following results show that summation from 2 to $\omega(G)$ shows better result.

Table 23: Full curriculum terms vs. curriculum learning with fewer terms

|  | IMDB | PTC-MM-$m$ | DSJC |
|---|---|---|---|
| MXNET | 0.094 | 0.179 | 0.383 |
| MXNET (Few-Curriculum) | 0.109 | 0.417 | 0.490 |

This is because, summation from 2 to $\omega(G)$ is providing more explicit guidance to the learner that, $K_2, .. K_c$ are subgraphs of $G$. Otherwise, it is unable to reason that if $K_c$ is a subgraph of $G$ then also $K_2, ..,$ are also subgraphs of $G$.

## F.11 COMPARISON WITH COMBINATORIAL METHODS

Here, we discuss several combinatorial methods used as baselines and compare their performance against MXNET.

**Fast Local Subgraph Counting (SCOPE)** SCOPE, proposed by (Li & Yu, 2024b) is a tree-decomposition-based method for local subgraph counting. It estimates the occurrence of a given query graph in the local neighborhood of each node within the corpus graph and provides a count for every corpus graph node. The method utilizes tree decomposition and symmetry-breaking rules to minimize redundancy and incorporates a novel multi-join algorithm for efficient computation. To predict the clique number, we supply SCOPE with a set of query clique graphs and retrieve the corresponding local subgraph counts. For every query clique smaller than the clique number, at least one node in the corpus graph contains the clique pattern in its local neighborhood. We determine clique presence by taking the maximum of the individual neighborhood counts, and the largest query clique with a non-zero count is deemed the predicted clique number.

It is important to note that SCOPE addresses a significantly more challenging task: estimating the count of patterns in each local neighborhood. As such, it naturally requires substantially more computational time compared to MXNET and earlier global subgraph counting methods. For each query–corpus graph pair provided to SCOPE, we set a timeout from the range [0.1, 1, 10, 100] seconds. If the timeout elapses without returning a result, we assume that no cliques were detected, and the clique number is determined accordingly.

Table 24: Comparison of SCOPE and MXNET across datasets (in terms of MSE) and timeout settings (in seconds). Numbers highlighted in green, and blue, indicate the best and second-best performers, respectively.

| Timeout (sec) | IMDB | AIDS-$m$ | PTC-MM-$m$ | DSJC | Brock | Enzymes | RB | MUTAG-$m$ |
|---|---|---|---|---|---|---|---|---|
| 0.1 | 42.620 | 6.140 | 1.486 | 274.735 | 595.545 | 0.613 | 185.525 | 68.565 |
| 1 | 25.000 | 0.005 | 0.000 | 226.165 | 469.120 | 0.000 | 84.975 | 50.255 |
| 10 | 23.148 | 0.005 | 0.000 | 217.180 | 430.180 | 0.000 | 81.160 | 32.175 |
| 100 | 20.093 | 0.000 | 0.000 | 192.615 | 398.050 | 0.000 | 68.040 | 12.515 |
| MXNET | 0.094 (0.029 sec) | 0.341 (0.021 sec) | 0.179 (0.018 sec) | 0.383 (0.014 sec) | 10.022 (0.014 sec) | 0.196 (0.023 sec) | 6.875 (0.119 sec) | 0.613 (0.134 sec) |

We observe that MXNET significantly outperforms SCOPE across all datasets when SCOPE is limited to a per-graph time of 0.1 seconds, which is comparable to the highest inference time taken

by MxNet. Furthermore, SCOPE's MSE consistently improves as it is afforded more computational time per graph, achieving perfect predictions on datasets such as Enzymes, AIDS-m, and PTC-MM-m. However, these improvements come at the cost of a substantial increase in inference time—exceeding 50x compared to MxNet.

Additionally, on several datasets, including IMDB-m, DSJC, Brock, RB, and MUTAG-m, SCOPE is unable to outperform MxNet even when given up to 100 seconds per graph, while MxNet achieves superior accuracy at a fraction of the computational cost. This behavior is expected, as SCOPE tackles a more challenging task of local subgraph counting, a level of granularity that is unnecessary for the specific task of clique number prediction. These results highlight MxNet's efficiency and suitability for practical applications requiring fast and accurate inference.

**Scaling Up $k$-Clique Densest Subgraph Detection (SCTL)** (He et al., 2023b) proposes SCTL to address the $k$-clique densest subgraph problem. To use SCTL for clique number detection, we initially employed the same protocol as MxNet, SCOPE, and NeurSC — incrementally increasing $k$ from 2 to $\omega(G)$. However, we found that when $k$ is smaller than $\omega(G)$, the increase in the number of $k$-cliques leads to prohibitively long runtimes — often hundreds of seconds per graph for a single $k$, which is far from the millisecond-level inference times achieved by MxNet and other baselines.

To optimize SCTL's runtime, we reversed the search direction: starting with the maximum possible clique size (equal to the graph size), we detect the absence of a clique (which is much faster) and decrease $k$ until a clique is detected. This adjustment significantly reduces computational overhead. However, SCTL is still 5x (PTC-MM-m, Brock) to 20x (AIDS-m, MUTAG-m) slower than MxNet.

**Comparison with upper and lower bounds on $\omega(G)$** We present results on different bound based methods as follows. We consider the best of the lower bounds and upper bounds as presented by the paper in terms of MSE, and present them here:

Table 25: Comparison with upper and lower bounds on $\omega(G)$

|  | Enzymes | PTC-MM-$m$ | RB |
|---|---|---|---|
| MxNet | 0.196 | 0.179 | 6.875 |
| Best Upper-Bound (UB) (Budinich, 2003, Eq. 1,2,3,4) | 1.756 | 637.70 | 532.530 |
| Best Lower-Bound (LB) (Budinich, 2003, Eq. 5,6,7) | 3.429 | 15.101 | 253.800 |

In particular, the bounds are:

1. **(UB) Eq. 1:** $\omega(G) \leq \frac{3+\sqrt{9-8(|V|-|E|)}}{2}$
2. Let $\rho(G)$ denote the spectral radius of the adjacency matrix $A$ of $G$. Then, **(UB) Eq. 2:** $\omega(G) \leq \rho(G) + 1$
3. Let $N_{-1} = |\{\lambda_i : \lambda_i \leq -1\}|$ where $\{\lambda_i\}_{i=1}^{|V|}$ denote the eigenvalues of $A$. Then , **(UB) Eq. 3:** $\omega(G) \leq N_{-1} + 1$
4. Let $\bar{A}$ denote the adjacency matrix of the complementary graph of $G$. Then, **(UB) Eq. 4:** $\omega(G) \leq |V| - \frac{\text{rank}(\bar{A})}{2}$
5. **(LB) Eq. 5:** $\omega(G) \geq \frac{1}{1 - \frac{2|E|}{|V|^2}}$
6. Let $\lambda_P$ denote the Perron eigenvalue of $A$, and $\boldsymbol{v}_P$ denote the corresponding eigenvector. Then, **(LB) Eq. 6:** $\omega(G) \geq \frac{\lambda_P}{(\mathbf{1}^\top \boldsymbol{v}_P)^2 - \lambda_P} + 1$
7. Let $\lambda_j$ denote an eigenvalue of $A$, and $\boldsymbol{v}_j$ denote the corresponding eigenvector. Define $g_j(\alpha) = \frac{\alpha^2 \lambda_j + (1-\alpha^2)\lambda_P}{\alpha(\mathbf{1}^\top \boldsymbol{v}_j) + \sqrt{1-\alpha^2}(\mathbf{1}^\top \boldsymbol{v}_P)}$.

   For each $j$, $g_j(\alpha)$ is defined on $\left[ \max_{i:\boldsymbol{v}_{ji}>0} \frac{-\boldsymbol{v}_{Pi}}{\sqrt{\boldsymbol{v}_{Pi}^2 + \boldsymbol{v}_{ji}^2}}, \min_{i:\boldsymbol{v}_{ji}<0} \frac{\boldsymbol{v}_{Pi}}{\sqrt{\boldsymbol{v}_{Pi}^2 + \boldsymbol{v}_{ji}^2}} \right]$. Then, for graphs which are not regular complete multipartite, **(LB) Eq. 7:** $\omega(G) \geq \frac{1}{1 - \max_{j,\alpha} g_j(\alpha)}$

