# OpenReview forum: "Clique Number Estimation via Differentiable Functions of Adjacency Matrix Permutations"
_ICLR.cc/2025/Conference — ICLR 2025 Poster_

### Official Review · Reviewer_U7Wn · 2024-10-30

**Soundness:** 3
**Presentation:** 3
**Contribution:** 3
**Rating:** 8
**Confidence:** 4

**Summary:**

This paper introduces MXNET, an end-to-end differentiable model for estimating the clique number in the graph. It contains MXNET(MSS) detection that detects cliques via message passing on permuted adjacency matrix and MXNET(SubMatch) that curriculum matches a series of cliques. Extensive experiments highlight the effectiveness of the proposed methods.

**Strengths:**

S1: It integrates both the message-passing-based clique detection and the matching-based techniques for accurate clique number estimation.

S2: The design of the proposed method has sufficient motivations.

S3: Experiments over multiple datasets validate its effectiveness.

**Weaknesses:**

W1: A more thorough comparison and introduction of related works would enhance the quality of this paper.
W1.1: A related field is subgraph counting [1]. We can directly use 2-cliques, 3-cliques, ..., k-cliques as query graphs and apply subgraph counting methods to determine the number of k-cliques in the data graph. If the number of k-cliques is 1, then the clique number may be k. Given the limited availability of direct baselines for the problem of clique number estimation, this method could serve as a promising baseline. Some promising methods are in [1] and [4]. The first one stands as a representative that uses the neural method to estimate the subgraph number while the second, while the second one stands as a representative that designs an algorithm for subgraph counting.

W1.2: Another promising area is the development of algorithm-based methods for approximate maximum clique detection or k-clique detection, as well as exact maximum clique detection [2]. These methods appear capable of quickly identifying accurate cliques. Exploring the motivation for using learning-based approaches and comparing these methods would be beneficial. Some promising methods are in [2] and [3]. The method in [2] is for exact maximum clique detection while the method in [3] is for approximate k-clique detection with theoretical guarantee which can be adapted for Clique Number Estimation.

========

W2: Some claims can be improved
W2.1: The authors seem to argue that relying on "extreme or no supervision" is not beneficial (Line 061).  However, intuitively, avoiding such supervision could enhance the generalization and applicability of the algorithms used. A better explanation would help the understanding of this paper.

W2.2: The author seems to focus on introducing prior work related to the Maximum Clique problem (Lines 49-62) in the introduction. However, the existing methods for estimating the clique number and their limitations are not sufficiently explored.

========

W3: Some experiments can be enhanced.

W3.1: As demonstrated in the related works introduced in Section 1.2, the clique number can be bounded. (There are also many other metrics that can be used to bound the clique number.) Given that the graph used in this paper is small and dense (see Table 7), employing these bounded metrics may yield good results and assist in pruning unpromising outcomes. As there are not many methods directly predicting the clique number, this bound-based baseline would be a good choice.

W3.2: The datasets evaluated in the paper are dense and small, with the largest containing only 300 nodes. It would be interesting to assess the performance on more diverse and larger datasets, as we can easily identify the exact cliques in smaller graphs. For example, the method proposed in previous works like in [3] can handle billion-scale graphs. Some smallest datasets in this paper like Email and Amazon would be helpful to asses the performance of the MXNET.



W3.3: The authors evaluate the effect of distribution shift; however, they remain within the same graph. It would be interesting to assess whether the trained model can generalize to different datasets, especially since there may be scenarios where labels are not available for training. An interesting scenario is that could the model trained in this paper can be generalized to predict the clique number for the datasets in [3].

[1]: "Neural Subgraph Counting with Wasserstein Estimator" in SIGMOD 2022

[2]: "Finding the Maximum Clique in Massive Graphs" in VLDB 2017

[3]: "Scaling Up 𝑘-Clique Densest Subgraph Detection" in SIGMOD 2023

[4]: "Fast Local Subgraph Counting" in VLDB 2024

**Questions:**

Q1: How do these subgraph counting algorithms perform in the clique number estimation?

Q2: How do these bound-based algorithms perform in the clique number estimation?

Q3: How does the proposed method perform in large graph.

---

> ### Author Response · Authors · 2024-11-20
> **Response to Reviewer U7Wn (1/3)**
>
> We thank the reviewer for their suggestions, which we discuss below and have incorporated in the paper.
>
> > *W1.1: A related field is subgraph counting [1,4].*
>
> We appreciate the reviewer’s suggestion regarding subgraph counting methods as potential baselines for our problem of clique number estimation.
>
> **Neural Subgraph Counting with Wasserstein Estimator (NeurSC) [1]**
>
> We adapt NeurSC ([1] cited by reviewer) as a baseline for our clique number prediction method by employing increasing $k$-cliques as query graphs and determining the subgraph count for each. The clique number is then estimated as the largest value of $k$ for which a non-zero count is returned.
> We present the results (MSE) of NeurSC in the following table, which shows that our method MxNet performs better.
>
> |Dataset|IMDB|AIDS-m|PTC-MM-m|DSJC|Brock|Enzymes|RB|MUTAG-m|
> |-|-|-|-|-|-|-|-|-|
> |NeurSC|5.33|1.24|1.51|15.35| 88.79|0.63|10.08|3.52|
> |MxNet|0.056|0.350|0.204|0.200|7.615|0.235|7.170|0.890|
>
> We have also incorporated NeurSC's results into Table 1 of the main paper for completeness. While our proposed MxNet method consistently outperforms NeurSC across all datasets, NeurSC demonstrates strong performance in certain scenarios. Specifically, it ranks as the third-best performer among decoder and non-decoder models for the Enzymes and RB datasets. Additionally, among non-decoder-based models, NeurSC ranks third for the PTC-MM-m, AIDS-m, and MUTAG-m datasets. These results underscore the utility of integrating neural modules to obtain clique presence certificates, which are effective for clique number estimation.
>
> We thank the reviewer for suggesting this highly relevant and competitive baseline, which has enriched our analysis and strengthened our comparisons.
>
> **Fast Local Subgraph Counting (SCOPE) [4]**
>
> SCOPE is a tree-decomposition-based method for local subgraph counting. It estimates the occurrence of a given query graph in the local neighborhood of each node within the corpus graph and provides a count for every corpus graph node. The method utilizes tree decomposition and symmetry-breaking rules to minimize redundancy and incorporates a novel multi-join algorithm for efficient computation. To predict the clique number, we supply SCOPE with a set of query clique graphs and retrieve the corresponding local subgraph counts. For every query clique smaller than the clique number, at least one node in the corpus graph contains the clique pattern in its local neighborhood. We determine clique presence by taking the maximum of the individual neighborhood counts, and the largest query clique with a non-zero count is deemed the predicted clique number.
>
> It is important to note that SCOPE addresses a significantly more challenging task: estimating the count of patterns in each local neighborhood. As such, it naturally requires substantially more computational time compared to MxNet and earlier global subgraph counting methods. For each query–corpus graph pair provided to SCOPE, we set a timeout from the range [0.1, 1, 10, 100] seconds. If the timeout elapses without returning a result, we assume that no cliques were detected, and the clique number is determined accordingly.
>
> The table below presents the performance of SCOPE under various timeout settings for each corpus graph. For MxNet, the numbers in parentheses indicate the average time taken per corpus graph (in seconds). We compare SCOPE against MxNet in terms of mean squared error (MSE).
>
> |Timeout (sec)| IMDB-m| AIDS-m| PTC _MM-m| DSJC| Brock| Enzymes| RB| MUTAG-m|
> |-|-|-|-|-|-|-|-|-|
> |0.1| 42.620| 6.140|1.486| 274.735| 595.545| 0.613| 185.525| 68.565|
> |1| 25.000| 0.005|0.000| 226.165| 469.120| 0.000| 84.975| 50.255|
> |10| 23.148|0.005|0.000| 217.180| 430.180| 0.000| 81.160| 32.175|
> |100| 20.093|0.000|0.000| 192.615| 398.050| 0.000| 68.040| 12.515|
> |MxNet| 0.056 (0.029 sec)| 0.350 (0.021 sec)| 0.204 (0.018 sec)| 0.200 (0.014 sec)| 7.615 (0.014 sec)| 0.235 (0.023 sec)| 7.170 (0.119 sec)| 0.890 (0.134 sec)|
>
> We observe MxNet significantly outperforms SCOPE across all datasets when SCOPE is limited to a per-graph time of 0.1 seconds, which is comparable to the highest inference time taken by MxNet. Also, SCOPE's MSE consistently improves as it is afforded more computational time per graph, achieving perfect predictions on datasets such as Enzymes, AIDS-m, and PTC-MM-m. However, these improvements come at the cost of a substantial increase in inference time—exceeding 50x compared to MxNet.
>
> Additionally, on several datasets, SCOPE is unable to outperform MxNet even when given up to 100 seconds per graph, while MxNet achieves superior accuracy at a fraction of the computational cost. This behavior is expected, as SCOPE tackles a more challenging task of local subgraph counting, a level of granularity that is unnecessary for the specific task of clique number prediction. These results highlight MxNet's efficiency and suitability for practical applications requiring fast and accurate inference.

---

> ### Author Response · Authors · 2024-11-20
> **Response to Reviewer U7Wn (2/3)**
>
> > *W 1.2. algorithms for approximate maximum clique detection or k-clique detection  [2,3].*
>
> **Scaling Up k-Clique Densest Subgraph Detection (SCTL) (paper [3] cited by the reviewer)**
>
> The SCTL paper addresses the  k-clique densest subgraph problem. To use SCTL for clique number detection, we initially employed the same protocol as MxNet, SCOPE, and NeurSC ― incrementally increasing k from 2 to $\omega(G)$. However, we found that when $k$ is smaller than $\omega(G)$, the increase in the number of $k$-cliques leads to prohibitively long runtimes ― often hundreds of seconds per graph for a single $k$, which is far from the millisecond-level inference times achieved by MxNet and other baselines.
>
> To optimize SCTL's runtime, we reversed the search direction: starting with the maximum possible clique size (equal to the graph size), we detect the absence of a clique (which is much faster) and decrease $k$ until a clique is detected. This adjustment significantly reduces computational overhead. However, SCTL is still 5x (PTC-MM-m, Brock) to 20x (AIDS-m, MUTAG-m) slower than MxNet.
>
> We could not find the code of "Finding the maximum clique in massive graphs, VLDB 2017".
>
> > *W2.1: The authors seem to argue that relying on "extreme or no supervision" is not beneficial (Line 061)... A better explanation would help the understanding of this paper.*
>
> We meant to state that existing methods fall in two broad regimes. Combinatorial methods often use no training at all, whereas methods from the ML community (e.g., Sun & Yang, 2024) depend on full supervision with a demonstration of an actual maximal clique. Our paper focuses on the neglected middle ground where supervision is provided as only the clique number (which can be estimated from various efficient relaxations) and not a clique demonstration (which is computationally expensive to arrange for training). In several graph search and retrieval scenarios, the relevance of a corpus graph depends on the extent to which query and corpus graphs overlap, which can be reduced to a maximum common induced subgraph (size) computation. In such cases, only clique number is available for supervision. We have updated the paper with further explanation.
>
> > *W2.2 existing methods for estimating the clique number are not sufficiently explored.*
>
> We were unable to find neural models that estimate clique number with less emphasis on maximum clique. But we discussed papers from discrete algorithms on clique number estimation without maximum clique computation in Section 1.2. We indeed thank the reviewer for the suggested papers from database domain, which are quite intriguing. We added those and some other papers which we found to be related in Appendix C, where we discussed related work in details.
>
>
> > *W3.1: the clique number can be bounded  bound-based baseline would be a good choice.*
>
> We consider the following three Lower-Bounds and four Upper-Bounds from "Exact bounds on the order of the maximum clique of a graph," Budinich. 2003, Discrete Applied Mathematics  (In Eq 7, Eq 4).
>
> In particular, the bounds are:
>
> 1. **(UB) Eq. 1**:
>    $$\omega(G) \leq \frac{3+\sqrt{9 - 8(|V| -|E|)}}{2}$$
>
> 2. Let $\rho(G)$ denote the spectral radius of the adjacency matrix $A$ of $G$. Then,
>    **(UB) Eq. 2**:
>    $$\omega(G) \leq \rho(G) + 1$$
>
> 3. Let $N _{-1} =|\{\lambda _i: \lambda _i \leq -1\}|$ where $(\lambda _i) _{i=1}^{|V|}$ denote the eigenvalues of $A$. Then,
>    **(UB) Eq. 3**:
>    $$\omega(G) \leq N _{-1} + 1$$
>
> 4. Let $\bar{A}$ denote the adjacency matrix of the complementary graph of $G$. Then,
>    **(UB) Eq. 4**:
>    $$\omega(G) \leq|V| - \frac{\text{rank}(\bar{A})}{2}$$
>
> 5. **(LB) Eq. 5**:
>    $$\omega(G) \geq \frac{1}{1 - \frac{2|E|}{|V|^2}}$$
>
> 6. Let $\lambda _P$ denote the Perron eigenvalue of $A$, and $\mathbf{v} _P$ denote the corresponding eigenvector. Then,
>    **(LB) Eq. 6**:
>    $$\omega(G) \geq \frac{\lambda _P}{(\mathbf{1}^\top \mathbf{v} _P)^2 - \lambda _P} + 1$$
>
> 7. Let $\lambda _j$ denote an eigenvalue of $A$, and $\mathbf{v} _j$ denote the corresponding eigenvector. Define
>    $$g _j(\alpha) = \frac{\alpha^2\lambda _j + (1-\alpha^2)\lambda _P}{\alpha (\mathbf{1}^\top \mathbf{v} _j) + \sqrt{1-\alpha^2}(\mathbf{1}^\top \mathbf{v} _P)}.$$
>    For each $j$, $g _j(\alpha)$ is defined on
>    $$\left[\max _{i: \mathbf{v} _{ji} > 0} \frac{-\mathbf{v} _{Pi}}{\sqrt{\mathbf{v} _{Pi}^2 + \mathbf{v} _{ji}^2}}, \min _{i: \mathbf{v} _{ji} < 0} \frac{\mathbf{v} _{Pi}}{\sqrt{\mathbf{v} _{Pi}^2 + \mathbf{v} _{ji}^2}}\right].$$
>    Then, for graphs which are not regular complete multipartite,
>    **(LB) Eq. 7**:
>    $$\omega(G) \geq \frac{1}{1 - \max _{j,\alpha} g _j(\alpha)}$$
>
> We report the *MSE* values of the best of the lower and best of the upper bounds as follows.
>
> |Method|Enzymes|PTC-mm|RB|
> |-|-|-|-|
> | MxNet| **0.235**| **0.204**| **7.170**|
> |Best-Lower-Bound|3.429|15.101|253.800|
> |Best-Upper-Bound|1.756|637.70|532.53|
>
> We observe that our method performs better.

---

> ### Author Response · Authors · 2024-11-20
> **Response to Reviewer U7Wn (3/3)**
>
> > *W3.2: [large datasets] Email and Amazon would be helpful to asses the performance of the MXNET.*
>
>
>
> There are two reasons why our primary focus was not on large graphs in the initial version.  However, we have now stress-tested our method with additional experiments, as described later.
>
>
> (1) Our goal is to design an *end-to-end differentiable model* ― which is guided by the needs of practical applications like estimation of size of maximum induced common subgraph for graph simialrity. These applications typically involves fewer than 200 nodes (but possibly very many graphs).
>
> Still, some of our datasets are much larger than the datasets used by the baselines. Specifically,  our method uses graphs with upto 33K edges (Mutag-m), which is larger  than all other datasets used in any end-to-end differentiable neural network based method. Moreover, we use the RB dataset, which is known to be notoriously hard for clique detection. Since clique number computation stands out as one of the hardest NP-complete problems [A,B], designing neural methods for even such moderate sized graphs is already quite challenging. It is unsurprising that neural methods have all limited themselves to relatively small graphs.
>
> (2) Like any other ML task, neural clique detectors need *multiple* graphs for training and one entire graph represents one instance of data. Data sets with a single large graph are more common than data sets containing multiple large graphs with known clique numbers, which prevents a neural method to test its performance on large graphs.  Nevertheless, we have the following proposal to predict clique number for a large graph, from which all neural models can benefit.
>
> We decompose a large graph $\mathcal{G}$ into overlapping subgraphs $G _1,..,G _N$  each with moderate sizes.  Then, the clique  number of $\mathcal{G}$ is estimated as $\max _i \omega(G _i)$.
> This decomposition is helpful because (1) ground truth clique number generation is much easier in smaller graphs and (2) a neural clique detector requires multiple (graph, clique number) pairs for training.  The risk of missing a clique straddling multiple subgraphs can be reduced via various degree based or K-core decomposition based heuristics.
>
>
> During the rebuttal, we worked with  Amazon and email-Enron datasets. Amazon has 334,863 nodes and 925,872 edges. email-Enron has 36,692 nodes and 183,831 edges. The table below shows the true and predicted clique numbers.
>
>
> ||  $\omega(G)$|  $\widehat{\omega}(G)$|
> |:-:|:-:|:-:|
> | Amazon| 7| 6|
> |email-Enron| 20| 18|
>
> We note that in each dataset, (1) MxNet was trained on a small set of subgraphs, sampled from the available single large graph, with corresponding ground truth clique numbers found using the Gurobi solver. During inference, MxNet effectively generalized to a much larger set of subgraphs extracted using degree-based heuristics, to ensure more comprehensive coverage of the large graph; (2) MxNet correctly identifies the subgraph containing the largest clique of the whole graph, i.e., $\arg \max _i \omega(G _i) = \arg \max _i \widehat{\omega}(G _i)$; (3) MxNet correctly predicts most of the member nodes of the maximum clique within the graph; (4) MxNet's predicted clique number for the large graph closely approximates the true clique number ― much better than the predictions made by baselines, even with smaller graphs.
>
> ---
>
> [A] Håstad, J. (1999), Clique is hard to approximate within $n^{1-\epsilon}$, Acta Mathematica.
>
> [B] Zuckerman, D. (2006), Linear degree extractors and the inapproximability of max clique and chromatic number, STOC.
>
>
> > *whether the trained model can generalize to different datasets*
>
>
> We train our model on AIDS-m and use it for testing on PTC-MM-m, to get an MSE of 0.802. When our model was trained on PTC-MM-m and tested on itself, we got an MSE of 0.204.
> Our approach approximates an NP-hard problem. To do so, we primarily exploite the distributional patterns present in the training data, enabling the model to specialize in those specific characteristics. Consequently, in out-of-distribution (OOD) experiments—where the training and testing datasets differ—the model's performance may degrade as it encounters patterns and distributions it has not been optimized to handle.

---

> ### Comment · Reviewer_U7Wn · 2024-11-25
>
> Thank you for your comments and the improved version of the paper. It has well addressed my concerns, and thus I update the score.

---

> ### Author Response · Authors · 2024-11-27
> **Thanks!**
>
> Thank you very much for your encouraging comments.

---

### Official Review · Reviewer_GJPR · 2024-11-04

**Soundness:** 3
**Presentation:** 2
**Contribution:** 3
**Rating:** 6
**Confidence:** 3

**Summary:**

In this work, the authors present a method for learning the maximum clique size in graphs by parameterizing node permutations using neural networks and training them with a composition of two types of losses, maximal subsquare (MSS) and SubMatch. Combined with techniques including various neural relaxations, curriculum training and bi-criteria early stopping, the model is able to learn to predict maximum clique size better than or close to the best among a number of baseline methods on several benchmark datasets.

**Strengths:**

While prior works have proposed various deep learning approaches for solving the maximum clique problem, the "distant supervision regime" where only the size of the maximum clique is provided during training is under-explored, and the authors' proposal to parameterize node permutation matrices with GNNs and Sinkhorm iterations is novel to my knowledge and an interesting one. The authors have also performed extensive numerical experiments to compare the proposed method with several baselines as well as analyzing the various design choices such as the loss choice and the early stopping criteria. The paper is clearly-written overall with technical details well-documented.

**Weaknesses:**

As acknowledged by the authors, the computational burden of the MxNet model may be a concern when dealing with larger graphs, and indeed the datasets used in the experiments consist of relatively small graphs.

See two questions below regarding some design choices in the MxNet model.

**Questions:**

1. In Section 3.1, the authors proposed a differentiable approximation of the combinatorial MCP by relaxing the discrete objective in three fronts, and I am curious if there is evidence or to believe that the first (relaxing the adjacency matrix to be continuously-valued embeddings) and the third (replacing Algorithm 1 by a message passing GNN) are necessary / beneficial in the case of featureless graphs. Without them, it seems that the model can still be trained via a differentiable loss function and the model might actual be more interpretable. A similar question on the relaxation of the clique average loss for the definition of MxNet (SubMatch) in Section 3.2.

2. In equation (8) for the subgraph matching problem, the last inequality ($K_{\omega(G)} \leq P A P^{\intercal}$) necessarily entails the earlier ones (e.g., $K_1 \leq P A P^{\intercal}$). Hence, instead of using the full curriculum with $c$ ranging from $2$ to $\omega(G)$, it seems possible to consider a simplification of the SubMatch loss in (9) by reducing the first (respectively, second) inner summation over $c$ to only the last term with $c = \omega(G)$ (respectively, $c = \omega(G)-1$). Is there evidence that including the full sequence of $c$'s is helpful?

Finally just a typo: On the bottom of Page 7, "$u$ indicating" --> "indicates".

---

> ### Author Response · Authors · 2024-11-20
> **Response to Reviewer GJPR (1/2)**
>
> We thank the reviewer for their suggestions, which we address as follows.  We have also added them in the revised version of the  paper.
>
> > *the computational burden of the MxNet model may be a concern when dealing with larger graphs*
>
> The limitation we stated applies not just to our method but for all neural methods. There are two reasons why our primary focus was not on large graphs in the initial version. However, we have now stress-tested our method with additional experiments, as described later.
>
> (1) Our goal is to design an *end-to-end differentiable model* ― which is guided by the needs of practical applications like estimation of size of maximum induced common subgraph for graph simialrity. These applications typically involves fewer than 200 nodes (but possibly very many graphs).
>
> Still, some of our datasets are much larger than the datasets used by the baselines. Specifically,  our method uses graphs with upto 33K edges (Mutag-m), which is larger  than all other datasets used in any end-to-end differentiable neural network based method. Moreover, we use the RB dataset, which is known to be notoriously hard for clique detection. Since clique number computation stands out as one of the hardest NP-complete problems [A,B], designing neural methods for even such moderate sized graphs is already quite challenging. It is unsurprising that neural methods have all limited themselves to relatively small graphs.
>
> (2) Like any other ML task, neural clique detectors need *multiple* graphs for training and one entire graph represents one instance of data. Data sets with a single large graph are more common than data sets containing multiple large graphs with known clique numbers, which prevents a neural method to test its performance on large graphs.  Nevertheless, we have the following proposal to predict clique number for a large graph, from which all neural models can benefit.
>
> We decompose a large graph $\mathcal{G}$ into overlapping subgraphs $G_1,..,G_N$  each with moderate sizes.  Then, the clique  number of $\mathcal{G}$ is estimated as $\max_i \omega(G_i)$.
> This decomposition is helpful because (1) ground truth clique number generation is much easier in smaller graphs and (2) a neural clique detector requires multiple (graph, clique number) pairs for training.  The risk of missing a clique straddling multiple subgraphs can be reduced via various degree based or K-core decomposition based heuristics.
>
>
> During the rebuttal, we worked with  Amazon and email-Enron datasets. Amazon has 334,863 nodes and 925,872 edges. email-Enron has 36,692 nodes and 183,831 edges. The table below shows the true and predicted clique numbers.
>
>
> ||  $\omega(G)$|  $\widehat{\omega}(G)$|
> |:-:|:-:|:-: |
> | Amazon | 7 | 6 |
> |email-Enron | 20 | 18 |
>
> We note that in each dataset, (1) MxNet was trained on a small set of subgraphs, sampled from the available single large graph, with corresponding ground truth clique numbers found using the Gurobi solver. During inference, MxNet effectively generalized to a much larger set of subgraphs extracted using degree-based heuristics, to ensure more comprehensive coverage of the large graph; (2) MxNet correctly identifies the subgraph containing the largest clique of the whole graph, i.e., $\arg \max_i \omega(G_i) = \arg \max_i \widehat{\omega}(G_i)$; (3) MxNet correctly predicts most of the member nodes of the maximum clique within the graph; (4) MxNet's predicted clique number for the large graph closely approximates the true clique number ― much better than the predictions made by baselines, even with smaller graphs.
>
> ---
>
> [A] Håstad, J. (1999), Clique is hard to approximate within $n^{1-\epsilon}$, Acta Mathematica, 182.
>
> [B] Zuckerman, D. (2006), Linear degree extractors and the inapproximability of max clique and chromatic number, STOC.
>
>
>
> > *differentiable approximation of the combinatorial MCP: Benefits of first and third approximation and submatch?*
>
>
> **(1) Benefits of relaxing binary adjacency matrix to continuous values:** In the following, we show the results of ablation study.
>
> |  | IMDB | PTC-MM-m | RB |
> |:---:|:---:|:---:|:---:|
> | MXNET | **0.056** | **0.204** | **7.170** |
> |Binary $A$ in MXNET | 1.851 | 2.364 | 10.500 |
>
>
> We observe that our smooth surrogate works better.
>
> Next, we discuss the rationale for our design, and plausible explanations for its success.
>
> * CLIQUE is a classical hard-to-approximate problem in the worst case, so the promise of neural methods is to take advantage of *distributions* from which graphs are generated in an application. A natural [1] way toward this goal is to train node representations $x_u$ that simultaneously predict the graph $A$ (via a generic GNN that aligns $x_u^\top x_v$ with $A[u,v]$), *and* make clique number prediction feasible (through a suitable task-customized network) ― in a graph containing a $k$-clique that includes nodes $u$ and $v$, they share the same $k-1$ node neighborhood, leading to a high $x_u^\top x_v$ value.

---

> ### Author Response · Authors · 2024-11-20
> **Response to Reviewer GJPR (2/2)**
>
> [Contd from above]
>
> * In MxNet (MSS), the goal is to find node permutation $S$ that brings all 1s in $S A S^\top$ into a dense submatrix. Achieving this requires moving the zeros to the right places.  When $S$ is relaxed to continuous values, non-zeros spread throughout $S A S^\top$, making it difficult to detect the MSS. Various thresholds or nonlinearities nay reduce the problem, but also attenuate gradient signals when $x_u^\top x_v$ is very high or very low.  Thus, $x_u^\top x_v$ scores serve as auxiliary signals for clique detection, but simply adding them to $A$ contributes much noise in the estimate of $\omega$.  In contrast, the form $S (A \odot XX^{\top}) S^\top$ is better at absorbing the signal from $\{x_u\}$ without succumbing to its noise.
>
> * As a further mechanistic discussion, suppose, instead of $A\odot XX^{\top}$, we use only the binary adjacency matrix $A$. Then, for each parameter $\theta\in \mathbb{R}$, the gradient term will involve computation of  $S A \frac{dS^{\top}}{d\theta} +  \frac{dS}{d\theta} A S^{\top}$. Since $S$ is computed using iterative normalization of rows and columns, starting with an exponential term, $\frac{dS}{d\theta}$ will contain terms such as $S_{ij} S_{i'j'}$ [2]. Thus, $S A \frac{dS^{\top}}{d\theta}$ contain terms of the form $S_{ij} S_{i'j'} S_{i''j''}$, with each $S_{.,.} \in [0,1]$. This leads to attenuation of gradient signals. If we keep $A\odot XX^{\top}$, then the gradient will have $S_{ij}S_{i'j'}$ $[\frac{d XX^T}{d\theta}]_{a,b}$, which will allow more significant gradient signals. Moreover, due to continuous adjacency entries,   $S _{ij} S _{i'j'} S _{i''j''}$ will now be replaced with   $S _{ij} S _{i'j'} S _{i''j''} X _{a,b} X _{a',b'}$, where $X$s can control the gradient signals better, leading to actual learning of clique number computation for a distribution over graphs.
>
> [1] In neural NLP, it is common to pretrain a language model for masked word prediction, and simultaneously train for a suite of common language tasks (e.g., FLAN-T5), or fine-tune them later. We follow the same strategy.
>
> [2] This can be obtained from the derivative of softmax probabilities $\frac{d}{do_k} e^{o_j} / \sum_i e^{o_i} = -(e^{o_j} / \sum_i e^{o_i}) (e^{o_k}/ \sum_i e^{o_i})$
>
> **(2) Relaxing Algorithm 1 using message passing GNN:** $MSS(B)$ can accurately compute clique only when $B$  has (close to) binary values. However, our computation of $SWS^T$ involves relaxation of binary values to real values in $(0,1)$ from $S$ and +ve/-ve from $W$, which makes the entries of $B[i,j]$  to be +ve/-ve real numbers. Further, multiplication by B[i,j] in line 7 in Algorithm 1 compounds this issue. To reduce these deleterious effects of fractional numbers on clique detection, several thresholding/clipping functions, such as Sigmoid, Tanh, ReLU, and ReLU6, may be applied, achieving varying degrees of success (we tried these). However, the best results were obtained by our final GNN-inspired solution, which could implement some form of local adaptive soft thresholding, as the message wavefront progressed. Note that this GNN is extremely lightweight, consisting of only 20 parameters.
>
> Below, we compare the results of the best non-GNN thresholding variant (ReLU1 = min(max(0,x), 1)) against the results of MSS with the GNN-based Algorithm 1 implementation.
>
> |  | IMDB | PTC-MM | RB |
> |:---:|:---:|:---:|:---:|
> | MXNET | **0.056** | **0.204** | **7.170** |
> |No GNN in Alg 1 | 0.935 | 0.386 | 7.626 |
>
> We observe that using a GNN based implementation for Algorithm 1 gives a performance improvement (lower error) across all datasets.
>
>
> **(3) Relaxing clique coverage loss in MxNet:**  $\min _P[K _c-PAP^{\top}] _+$ is a
> quadratic assignment problem (QAP), which is a hard problem to solve. Moreover, the binary values of $A$ and $K _c$ attenuate the gradient signals as described in the last para of item (1) above. We overcome the above challenges by replacing  $\min_P[K _c-PAP^{\top}] _+$ with $[Q _c-SX] _+$. Instead of solving QAP, it solves a linear assignment problem, which is more tractable; and allows substantial gradient signals.
>
> || IMDB | PTC-MM | RB |
> |:-:|:-:|:-:|:-:|
> | MXNET | **0.056** | **0.204** | **7.170** |
> |Hard Clique Loss | 0.648 | 0.425 | 9.675 |
> > *[curriculum learning] is full sequence of c is helpful?*
>
> This is indeed a great suggestion. We performed experiments where in Eq (9), we limited the first inner summation to only $c= \omega(G)$ and the second summation to only $c=\omega(G)-1$. The following results show that summation from 2 to $\omega(G)$ shows better result.
>
> ||IMDB | PTC-MM | RBG |
> |-|-|-|-|
> |Full curriculum|**0.056** |**0.204** | **7.170**  |
> |Only last term|0.481| 1.339| 10.500|
>
>
>
> This is because, summation from 2 to $\omega (G)$ is providing more explicit guidance to the learner that, $K_2,..K_c$ are subgraphs of $G$. Otherwise, it is unable to reason that if $K_c$ is a subgraph of $G$ then also $K_2,..,$ are also subgraphs of $G$.

---

> > ### Comment · Reviewer_GJPR · 2024-11-27
> > **Thank you for the response**
> >
> > I really appreciate the response from the authors and the additional experiments, which are helpful in addressing my concerns and added clarifications on the motivation for the model design. I am willing to raise my score to '7'.

---

### Official Review · Reviewer_7fYN · 2024-11-08

**Soundness:** 3
**Presentation:** 3
**Contribution:** 3
**Rating:** 6
**Confidence:** 2

**Summary:**

In this submission, the authors propose a differentiable solution to the clique number estimation problem. In particular, they formulate the maximum clique problem (MCP) as a maximization of the size of a fully dense square submatrix within a suitably row-column-permuted adjacency matrix.
This problem can be further treated as a sequence of subgraph matching tasks, and the cliques can be progressively detected, leading to a fully differentiable neural network called MxNet. Experiments show that the proposed method outperforms baselines in most situations, which is more robust to the OOD issue.

**Strengths:**

1. The problem is significant for many applications.
2. The proposed method is reasonable, and the re-formulation of MCP is insightful.
3. The experiments are solid and sufficient. Analytic experiments such as ablation studies are provided.

**Weaknesses:**

1. The writing and organization of the proposed method are not friendly to readers without sufficient background. In the introduction section, adding a figure illustrating the task may help readers quickly grasp the key concepts of the paper. Similarly, in Figure 1, binding notations like MSS(B) and rho_theta to the modules in the figure is necessary.
2. The motivation and rationale behind certain modeling strategies are not clearly explained. For instance, in Figure 1, what is the purpose of using a smooth surrogate for the adjacency matrix? Is there an ablation study that supports this approach? Additionally, the effects of the bicriteria early stopping strategy have not been verified.
3. The scalability of the proposed method for large graphs has not been verified. In particular, the Sinkhorn network often experiences numerical instability when dealing with very large graphs. What approaches can be taken to mitigate this issue? Additionally, how should the temperature of the Gumbel-softmax be set within the network?
4. The lambda balances the two terms in the loss function, is the proposed method robust to this hyperparameter?

**Questions:**

Please see above.

---

> ### Author Response · Authors · 2024-11-20
> **Response to Reviewer 7fYN (1/2)**
>
> We thank the reviewer for their feedback. We address them here and have also added them in the revised version of the  paper.
>
> > *adding a figure illustrating the task*
>
> Acting on your suggestions, we have overhauled Figure 1, clearly showing how maximal dense subsquare detection works, and then illustrate the nested tests for a series of sub-clique containment tests, and finally the  overall differentiable composite loss. The caption has been modified to be clearer and more descriptive.
>
> > *purpose of  smooth surrogate for the adjacency matrix? ... ablation study*
>
> In the following, we show the MSE from ablation study, which shows smooth surrogate works better (smaller MSE).
> ||IMDB | PTC-MM-m | RB |
> |:-:|:-:|:-:|:-:|
> | MXNET | **0.056** | **0.204** | **7.170** |
> |Binary $A$ in MXNET | 1.851 | 2.364 | 10.500 |
>
> Next, we discuss the rationale for our design, and plausible explanations for its success.
>
> * CLIQUE is a classical hard-to-approximate problem in the worst case, so the promise of neural methods is to take advantage of *distributions* from which graphs are generated in an application. A natural [1] way toward this goal is to train node representations $x_u$ that simultaneously predict the graph $A$ (via a generic GNN that aligns $x_u^\top x_v$ with $A[u,v]$), *and* make clique number prediction feasible (through a suitable task-customized network) ― in a graph containing a $k$-clique that includes nodes $u$ and $v$, they share the same $k-1$ node neighborhood, leading to a high $x_u^\top x_v$ value.
> * In MxNet (MSS), the goal is to find node permutation $S$ that brings all 1s in $S A S^\top$ into a dense submatrix. Achieving this requires moving the zeros to the right places.  When $S$ is relaxed to continuous values, non-zeros spread throughout $S A S^\top$, making it difficult to detect the MSS. Various thresholds or nonlinearities nay reduce the problem, but also attenuate gradient signals when $x_u^\top x_v$ is very high or very low.  Thus, $x_u^\top x_v$ scores serve as auxiliary signals for clique detection, but simply adding them to $A$ contributes much noise in the estimate of $\omega$.  In contrast, the form $S (A \odot XX^{\top}) S^\top$ is better at absorbing the signal from $\{x_u\}$ without succumbing to its noise.
> * As a further mechanistic discussion, suppose, instead of $A\odot XX^{\top}$, we use only the binary adjacency matrix $A$. Then, for each parameter $\theta\in \mathbb{R}$, the gradient term will involve computation of  $S A \frac{dS^{\top}}{d\theta} +  \frac{dS}{d\theta} A S^{\top}$. Since $S$ is computed using iterative normalization of rows and columns, starting with an exponential term, $\frac{dS}{d\theta}$ will contain terms such as $S_{ij} S_{i'j'}$ [2]. Thus, $S A \frac{dS^{\top}}{d\theta}$ contain terms of the form $S_{ij} S_{i'j'} S_{i''j''}$, with each $S_{.,.} \in [0,1]$. This leads to attenuation of gradient signals. If we keep $A\odot XX^{\top}$, then the gradient will have $S_{ij}S_{i'j'}$ $[\frac{d XX^T}{d\theta}]_{a,b}$, which will allow more significant gradient signals. Moreover, due to continuous adjacency entries,   $S _{ij} S _{i'j'} S _{i''j''}$ will now be replaced with   $S _{ij} S _{i'j'} S _{i''j''} X _{a,b} X _{a',b'}$, where $X$s can control the gradient signals better, leading to actual learning of clique number computation for a distribution over graphs.
>
> [1] In neural NLP, it is common to pretrain a language model for masked word prediction, and simultaneously train for a suite of common language tasks (e.g., FLAN-T5), or fine-tune them later. We follow the same strategy.
>
> [2] This can be obtained from the derivative of softmax probabilities $\frac{d}{do_k} e^{o_j} / \sum_i e^{o_i} = -(e^{o_j} / \sum_i e^{o_i}) (e^{o_k}/ \sum_i e^{o_i})$
>
> > *bicriteria early stopping strategy have not been verified.*
>
> We found bicriteria early stopping to be an effective guardrail against potential overfitting of MXNET (MSS).  Below are the numbers for Brock and Enzyme. MSS based early stopping led to overfitting of MSS, which is why the performance of submatch was worse. But Bicriteria early stopping improved performance of submatch significantly, with a small increase in MSE of MSS. For Enzyme, Bicriteria early stopping improved the performance of submatch without any deterioration of  MSS.
>
> | | MSS | SubMatch | MSS  |SubMatch
> |-|-|-|-|-|
> |  |Brock | Brock |Enzyme|Enzyme |
> | ES=MSS | **7.615** | 1400.00| **0.235** | 0.328 |
> | ES=SubMatch | 21.265| **227.91** | **0.235** | 0.353 |
> | ES = Bicriteria | 9.260 | 307.90 | **0.235** | **0.261** |

---

> > ### Author Response · Authors · 2024-11-20
> > **Response to Reviewer 7fYN (2/2)**
> >
> > > *scalability for large graphs has not been verified*
> >
> > There are two reasons why our primary focus was not on large graphs in the initial version.  However, we have now stress-tested our method with additional experiments, as described later.
> >
> >
> > (1) Our goal is to design an *end-to-end differentiable model* ― which is guided by the needs of practical applications like estimation of size of maximum induced common subgraph for graph simialrity. These applications typically involves fewer than 200 nodes (but possibly very many graphs).
> >
> > Still, some of our datasets are much larger than the datasets used by the baselines. Specifically,  our method uses graphs with upto 33K edges (Mutag-m), which is larger  than all other datasets used in any end-to-end differentiable neural network based method. Moreover, we use the RB dataset, which is known to be notoriously hard for clique detection. Since clique number computation stands out as one of the hardest NP-complete problems [A,B], designing neural methods for even such moderate sized graphs is already quite challenging. It is unsurprising that neural methods have all limited themselves to relatively small graphs.
> >
> > (2) Like any other ML task, neural clique detectors need *multiple* graphs for training and one entire graph represents one instance of data. Data sets with a single large graph are more common than data sets containing multiple large graphs with known clique numbers, which prevents a neural method to test its performance on large graphs.  Nevertheless, we have the following proposal to predict clique number for a large graph, from which all neural models can benefit.
> >
> > We decompose a large graph $\mathcal{G}$ into overlapping subgraphs $G_1,..,G_N$  each with moderate sizes.  Then, the clique  number of $\mathcal{G}$ is estimated as $\max_i \omega(G_i)$.
> > This decomposition is helpful because (1) ground truth clique number generation is much easier in smaller graphs and (2) a neural clique detector requires multiple (graph, clique number) pairs for training.  The risk of missing a clique straddling multiple subgraphs can be reduced via various degree based or K-core decomposition based heuristics.
> >
> >
> > During the rebuttal, we worked with  Amazon and email-Enron datasets. Amazon has 334,863 nodes and 925,872 edges. email-Enron has 36,692 nodes and 183,831 edges. The table below shows the true and predicted clique numbers.
> >
> >
> > ||  $\omega(G)$|  $\widehat{\omega}(G)$|
> > |:-:|:-:|:-: |
> > | Amazon | 7 | 6 |
> > |email-Enron | 20 | 18 |
> >
> > We note that in each dataset, (1) MxNet was trained on a small set of subgraphs, sampled from the available single large graph, with corresponding ground truth clique numbers found using the Gurobi solver. During inference, MxNet effectively generalized to a much larger set of subgraphs extracted using degree-based heuristics, to ensure more comprehensive coverage of the large graph; (2) MxNet correctly identifies the subgraph containing the largest clique of the whole graph, i.e., $\arg \max_i \omega(G_i) = \arg \max_i \widehat{\omega}(G_i)$; (3) MxNet correctly predicts most of the member nodes of the maximum clique within the graph; (4) MxNet's predicted clique number for the large graph closely approximates the true clique number ― much better than the predictions made by baselines, even with smaller graphs.
> >
> > ---
> >
> > [A] Håstad, J. (1999), Clique is hard to approximate within $n^{1-\epsilon}$, Acta Mathematica, 182.
> >
> > [B] Zuckerman, D. (2006), Linear degree extractors and the inapproximability of max clique and chromatic number, STOC.
> >
> > > *How should temperature of Gumbel Sinkhorn (GS) should be set*
> >
> >
> > We found that the performance (MSE) is robust to the temperature of GS network as shown by the following result for PTC-MM.
> >
> >  | $\tau\to$|0.001|0.01|0.025|0.05|0.1|0.25|1 |
> > |-|-|-|-|-|-|-|-|
> > | MSE $\to$|0.237|0.238|0.200|0.204|0.293|0.276|1.796|
> >
> >
> > If the number of nodes $|V|$ is large, the entries of doubly stochastic matrix $S$ become more diffused. For the same temperature, the highest value in each row/column of $S$ becomes smaller as we increase $|V|$. Hence, we can decrease the temperature to ensure that $S$ has sufficiently low entropy to be interpretable.  Further, we can use reparameterization of weight matrices [C] in Sinkhorn network using spectral norm based scaling to facililate training in large graph training.
> >
> > [C] Stabilizing Transformer Training by Preventing Attention Entropy Collapse. Zhai et al. ICML 2023.
> >
> > > *The lambda balances the two terms in the loss function, is the proposed method robust to this hyperparameter?*
> >
> > Our experiments show that the proposed method is robust to the choice of $\lambda \in [0.5,2]$, as shown below (PTC-mm).
> >
> > | $\lambda \to$ | 0.1 | 0.25 | 0.5 | 1 | 2 | 5 | 10 |
> > | - | - | - | - | - | - | - | - |
> > | MSE $\to$ | 2.364 | 0.317 | 0.257 | 0.204 | 0.176 | 0.442 | 0.446 |

---

### Meta-Review · Area_Chair_GoHj · 2024-12-11

**Metareview:**

This paper consider estimating clique number through the lens of learning. The main technical ingredients are: reformulating the problem of finding fully dense square submatrix, solving a sequence of subgraph matching problems. Reviewers agree that the problem is important and the proposed approach is new. Reviewers yet also have concerns on the computational cost for large-scale problems. Since the main contribution of the paper is on algorithmic aspect (i.e. introducing a differentiable optimization approach), which reviewers were convinced of, the AC believes that the work is above the bar.

**Additional Comments On Reviewer Discussion:**

Despite some clarity questions, a primary concern from the reviewers is the computational cost. Authors responded that some data sets they used are already larger than in prior works, and that the key contribution is the introduction of a new differentiable algorithm. The AC and reviewers were convinced. The AC-reviewer discussion was not initiated due to the fairly clear scores.

---

### Decision · Program_Chairs · 2025-01-22

Accept (Poster)